# CableInspect-AD: An Expert-Annotated Anomaly Detection Dataset

**Akshatha Arodi**[1*]    **Margaux Luck**[1*]    **Jean-Luc Bedwani**[2]
**Aldo Zaimi**[1]    **Ge Li**[1]    **Nicolas Pouliot**[2]    **Julien Beaudry**[2]    **Gaétan Marceau Caron**[1]
[1]Mila - Quebec AI Institute    [2]IREQ - Institut de recherche d'Hydro-Québec
*equal contributions

## Abstract

Machine learning models are increasingly being deployed in real-world contexts. However, systematic studies on their transferability to specific and critical applications are underrepresented in the research literature. An important example is visual anomaly detection (VAD) for robotic power line inspection. While existing VAD methods perform well in controlled environments, real-world scenarios present diverse and unexpected anomalies that current datasets fail to capture. To address this gap, we introduce *CableInspect-AD*, a high-quality, publicly available dataset created and annotated by domain experts from Hydro-Québec, a Canadian public utility. This dataset includes high-resolution images with challenging real-world anomalies, covering defects with varying severity levels. To address the challenges of collecting diverse anomalous and nominal examples for setting a detection threshold, we propose an enhancement to the celebrated PatchCore algorithm. This enhancement enables its use in scenarios with limited labeled data. We also present a comprehensive evaluation protocol based on cross-validation to assess models' performances. We evaluate our *Enhanced-PatchCore* for few-shot and many-shot detection, and Vision-Language Models for zero-shot detection. While promising, these models struggle to detect all anomalies, highlighting the dataset's value as a challenging benchmark for the broader research community. Project page: `https://mila-iqia.github.io/cableinspect-ad/`.

## 1 Introduction

Machine learning is increasingly applied across diverse industrial fields such as robotics, genomics, climate and materials science due to the impressive performance of large pre-trained models. As the community looks towards deploying these models in specialized domains where their effectiveness remains uncertain, there is a pressing need to improve their transferability in these contexts. This underscores the necessity for tailored datasets by domain experts. Visual anomaly detection (VAD) in a specific industrial context, exemplifies a critical application, promising cost reduction, time savings, and enhanced safety measures by enabling preventive maintenance. While existing VAD methods perform well in controlled environments, real-world scenarios present diverse and unexpected anomalies that current datasets fail to capture. Public VAD datasets, such as MvTec AD [7], VisA [56], and MVTec LOCO AD [6], focus mainly on objects and textures in a controlled manufacturing context, thus limiting the scope of potential anomalies. Moreover, these datasets do not account for scenarios with significant variations of the same object, further complicating AD in real-world applications. For instance, objects may exhibit substantial differences when viewed indoors versus outdoors due to varying operational conditions and environmental factors such as lighting and weather. Additionally, wear and tear over time can introduce anomalies that evolve, creating multiple views and states of the same object. Compounding the complexity, images may contain more than one anomaly, requiring models to discern and identify multiple issues simultaneously.

38th Conference on Neural Information Processing Systems (NeurIPS 2024) Track on Datasets and Benchmarks.

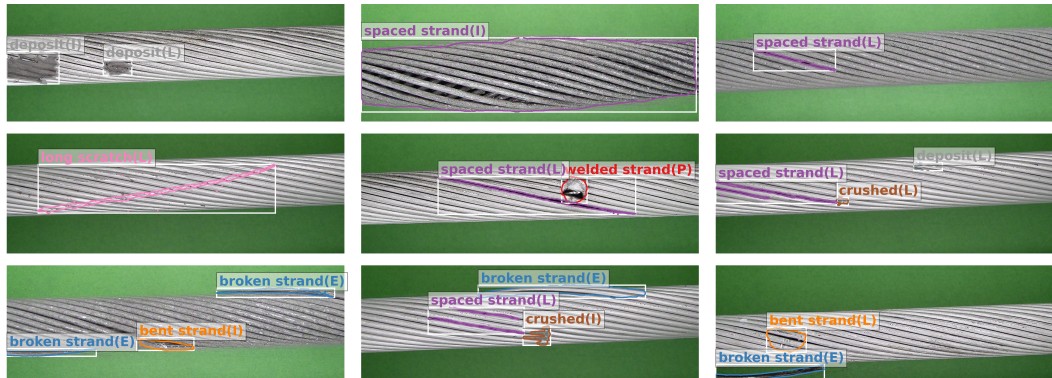

Figure 1: Examples of anomalies. On each image, the anomaly types (grades) are annotated (masks outlined). The grades here are (I)mportant, (L)ight, (C)omplete, (E)xtracted and (D)eep. Anomalies such as long scratches(I) are hard to spot, whereas deposit(I) and spaced strands(I) are easier.

Robotic power line inspection represents a specialized and highly challenging domain characterized by a wide range of anomalies, further complicated by the changing appearance of cables due to natural wear. Recognizing the importance of open-science and transparency in evaluating machine learning models for such complex real-world applications, there is a clear need for more public industrial datasets. To this end, we introduce *CableInspect-AD* (see Figure 1), a high-quality, publicly available dataset created and annotated by domain experts from Hydro-Québec[1], a Canadian public utility. It features 4,798 high-resolution images and 6,023 annotated anomalies across three types of power line cables. These anomalies represent the seven most common defect types listed by Hydro-Québec, with varying severity levels. They are meticulously crafted by experts and are annotated at the image level, the pixel-level, and with bounding boxes, to provide a detailed categorization of those anomalies both by type and by severity level.

To address the challenges of collecting diverse anomalous and nominal examples for setting a detection threshold, we introduce *Enhanced-PatchCore*, an improved approach that sets thresholds using only a training set with a few nominal images. This approach enhances adaptability and robustness to diverse anomaly types encountered in real-world industrial settings. Furthermore, this enhancement enables its application in scenarios with limited labeled data. We define a comprehensive evaluation protocol based on cross-validation and evaluate *Enhanced-PatchCore* for few-shot and many-shot detection. To further eliminate the need for a train set, we seek to use open[2] conversational Vision-Language Models (VLMs) [47, 28], which have demonstrated impressive capacity in zero-shot settings on tasks like Visual Question Answering (VQA) and image captioning. To the best of our knowledge, this is among the first attempts to utilize open conversational VLMs for zero-shot VAD in this context. Our findings indicate that the baselines show promising results in detecting anomalies on the cables. However, they struggle with certain types and grades of anomalies, highlighting the need for further research in real-world industrial contexts. By introducing *CableInspect-AD*, we aim to push the frontiers of VAD and demonstrate its potential to generalize to complex, real-world domains.

## 2   Related work

**Datasets**   Industrial VAD datasets such as MvTec AD [7], VisA [56] and MVTec LOCO AD [6] are commonly used to evaluate VAD methods, focusing primarily on objects and textures inspection in a controlled environment. However, despite their popularity, the evaluation results from these datasets may not translate to specialized domains, such as VAD in power line cables. Specifically, VisA excludes cables, and while MVTec AD does feature a cable category, it focuses on the cross-sectional aspect of cables and lacks the nuanced defects found in power line cables.

Existing public power line inspection datasets predominantly focus on specific power line components like transmission towers and insulators [33, 41, 42, 2, 44, 15, 8] and often overlook intricacies and

---

[1]https://en.wikipedia.org/wiki/Hydro-Qu%C3%A9bec
[2]Open models here are defined as those with widely accessible weights.

anomalies on cables. While the InsPLAD dataset [45] addresses both object detection (InsPLAD-det) and VAD (InsPLAD-fault), it does not feature anomalies on cables. In contrast, some datasets focus solely on power line cables but are primarily intended for cable detection or segmentation rather than inspection [9, 14, 23, 31, 1, 50]. For example, datasets designed for aircraft safety [50] or autonomous flying vehicles [14, 31] offer low-resolution, birds-eye-view shots of cables without annotations for cable anomalies. This lack of specialized datasets tailored to power line cable anomalies underscores the need for a new dataset.

**Anomaly detection algorithms**    VAD in industrial settings predominantly relies on *unsupervised* methods [30]. This preference mainly stems from the ease of obtaining nominal examples compared to the expensive and complex task of specifying expected defect variations.

Consequently, the training set often contains only nominal samples, while the validation and test sets include both anomalous and nominal samples for model evaluation. The taxonomy proposed by [30] classifies these methods into two broad categories: *reconstruction-based* and *feature-embedding-based* approaches.

Reconstruction-based approaches [53, 4, 24, 39, 49, 40, 52, 48] typically involve training encoder-decoder models. During testing, they predict anomalies by comparing the input image with its reconstruction, assuming models will generate errors for anomalies not part of the training set.

Feature-embedding-based approaches  [46, 25, 37, 43, 36] on the other hand, employ pre-trained models to generate embeddings for VAD. Among several methods, instance-based approaches are the most effective [30]. These methods store normal feature embeddings in a *memory bank*, where embeddings far from those in the memory bank are likely anomalous. Notably, the PatchCore [36] algorithm demonstrates significant advancements, achieving state-of-the-art results on benchmarks such as MVTec AD and VisA, showing promising performance in both few-shot and many-shot settings [38]. While methods like PatchCore can work with few nominal examples, they still need a comprehensive set of both nominal and anomalous images to select a threshold, which is impractical in real-world applications where collecting diverse anomalies is difficult. Consequently, these methods often face challenges in generalization, particularly when anomalies are rare and the nominal images are diverse, leading to unreliable performance.

More recent research has explored the application of large models and VLMs to VAD. Models based on CLIP [16, 21, 13], SAM and GroundingDINO  [10, 22], and conversational VLMs  [32, 11, 54, 18] have shown promising results. These models leverage the capabilities of VLMs in zero-/few-shot inference and image understanding tasks. For instance, [11, 54] demonstrates the potential of GPT-4V's generic capacity on zero-shot VAD tasks. However, it leverages a proprietary model with limited API access. In addition, AnomalyGPT [18], a conversational VLM fine-tuned for VAD tasks, requires finetuning on a set of nominal and simulated anomalous images, which can be costly and impractical in real-world VAD applications. In contrast, our study explores the use of open conversational VLMs for zero-shot VAD to ensure our comparisons are accessible and replicable within the research community.

Other recent works on utilizing large models for VAD tasks such as MuSc [26] and APRIL-GAN [12], while demonstrating competitive performances in zero-/few-shot scenarios, do not align with the approach opted in this work. Although MuSc is claimed to be a zero-shot method, it still relies on prior knowledge from a test set, a requirement that is impractical for real-world applications like power line inspection. In addition, the method assumes the test set contains abundant information on both normal and abnormal cues, which is not applicable in settings where only nominal images are available. APRIL-GAN, while achieving good results in certain contexts, requires an additional training phase, which is resource-intensive for training and evaluation on our dataset. Furthermore, WinCLIP [21] either matches or outperforms APRIL-GAN in similar contexts, making it a more suitable candidate for initial benchmarking.

## 3   CableInspect-AD dataset

Advances in robotics, exemplified by Hydro-Québec's LineRanger robot [35], have transformed power line inspections, introducing automation for increased efficiency [5, 19, 35]. Our *CableInspect-AD* dataset, developed by Hydro-Québec experts, plays a crucial role in furthering robotics through deep learning and serves as a benchmark for developing and evaluating new VAD algorithms with

real-world data. It addresses the challenge of detecting rare multi-scale anomalies on power line cables, which vary in wear, color, texture, and braiding. It also facilitates the extension of these techniques to other infrastructure-monitoring areas, such as railways and pipelines, fostering the evaluation of VAD models and the creation of predictive maintenance systems to advance VAD technologies across various sectors.

**Creation and annotation** The creation and annotation of *CableInspect-AD* is highly challenging and requires domain expertise. To achieve this, experts selected three cables used in the field. The cables are suspended for image acquisition, and an apparatus with a moving camera is used to capture the images to ensure a uniform background and mimic real-world robotic scenarios. The uniform background was intentionally chosen to minimize distractions and external factors, allowing models to focus solely on detecting anomalies within the object, a practice commonly seen in other VAD benchmarks. Importantly, capturing images while the apparatus is in motion introduces slight disturbances, making the images less perfect compared to datasets like MVTec AD, thereby adding to the dataset's uniqueness and realism. To maximize the use of each cable, both sides (referred to as sides A and B) are utilized.

For each cable side, three videos are recorded at a frame rate of 30 frames per second, consisting of RGBA images at a resolution of 1920×1080 pixels. A total of 18 videos are captured by manually moving a camera along the cables at different speeds, slow enough to capture a defect in several frames. Each pass includes minor rotational variations, up to 20 degrees, and can be taken forward or backward, slightly changing the perspective. The videos are then processed to keep one frame out of three for anomaly annotations, reducing the frame rate to 10 frames per second.

Annotations include image-level labels and bounding boxes, assigned based on expert assessment of the anomaly's appearance in the image. Additionally, per-pixel labels for the first recorded video on each cable are obtained using SAM [22] prompted with the bounding boxes and then manually corrected. Depending on the point of view, a defect can be associated with different grades. An image containing at least one bounding box is considered anomalous. Examples of anomalies are shown in Figure 1, illustrating their varying appearance and complexity. The dataset was annotated by at least four experts who first developed and agreed on guidelines to establish a clear annotation framework. The process was repeated five times until an agreement was achieved. The acquisition process, annotation guide, and details on the annotation process are in Supplementary Material.

**Statistics** The dataset contains 4,798 annotated images (2,639 anomalous and 2,159 nominal). Among the anomalous images, there are 193 unique anomalies, comprising 110 manually created and 83 pre-existing real-world anomalies. The total number of anomalies annotated is 6,023. The distribution of defects among the three cables is shown in Figure 2.

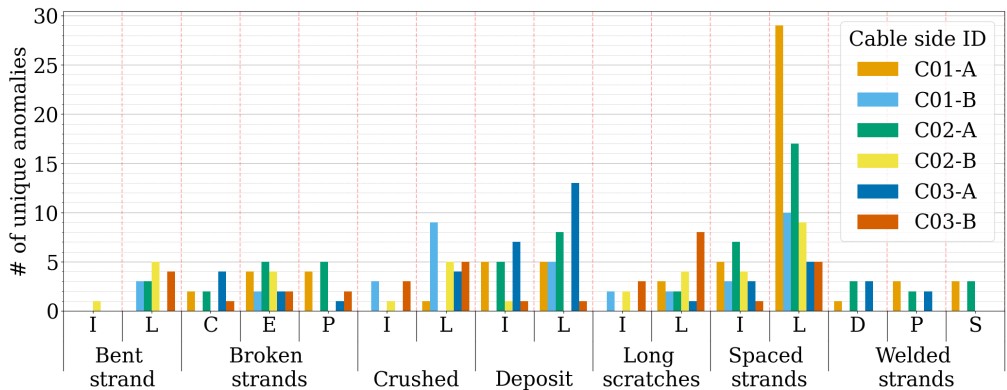

Figure 2: Anomaly types and grades per cable. The grades are (I)mportant, (L)ight, (C)omplete, (E)xtracted, (P)artial, (D)eep and (S)uperficial. The anomalies are not distributed uniformly across all the cables.

**Evaluation protocol** To estimate variance in model performance, we use a k-fold cross-validation strategy tailored to our dataset. This approach addresses the high anomaly ratio resulting from the

deliberate creation of diverse anomalies, the non-uniform distribution of anomalies, and possible data leakage due to overlapping video frames. Specifically, we split the power line cable dataset into train and test sets using a k-fold sampling strategy based on defect identifiers. For each fold, defect identifiers are randomly selected, and 100 subsequent nominal images are selected for training while preventing overlap between training and test sets using buffers. This process is repeated k times, ensuring a consistent training size but varying test images and anomaly ratios across folds as shown in Figure 3. More details can be found in the Supplementary Material.

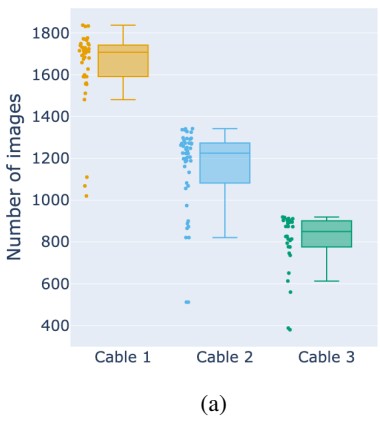

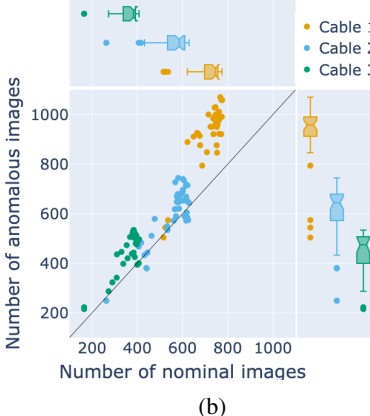

(a)

(b)

Figure 3: The three cables have different numbers of images with varying anomaly ratios in the test set. The cables have 40, 46, and 30 folds, respectively. (a) shows the number of images in the test set over all the folds for each cable (x-axis), and (b) shows the ratio in the test set of the cables. Each point corresponds to the anomaly ratio in a fold. The identity line shows where a balanced dataset would be.

## 4 Enhanced-PatchCore

*Enhanced-PatchCore*, built on PatchCore [36], is an instance-based approach that stores feature embeddings of nominal images in a *memory bank* $\mathcal{M}$ to establish a context during training. This memory bank is then coreset-subsampled [3] to reduce its size.

At test time, the abnormality of a test image $X$ is determined by measuring its distance to the nearest neighbor in the memory bank within the embedding space. This distance, referred to as anomaly score, is defined as:

$$S(X) := \max_{e \in \mathcal{P}(X)} d(e, \mathcal{M}) = \max_{e \in \mathcal{P}(X)} \min_{e' \in \mathcal{M}} d(e, e') \quad (1)$$

where $\mathcal{P}(X)$ is the set of patch embeddings generated by an image encoder and $d$ is the Euclidean distance.

To decide if an image contains an anomaly from this score, a threshold must be set using a validation set. However, creating a robust validation set with a diverse range of anomalies is prohibitively expensive. Many VAD methods overlook this crucial aspect, either manually setting thresholds or reporting the best F1 score. This is impractical in real-world applications, where thresholds must be carefully calibrated to specific operational requirements and constraints. Therefore, we introduce *Enhanced-PatchCore*, which addresses this challenge by setting a threshold using only the train set. Specifically, it computes anomaly scores of images within the memory bank to estimate the empirical distribution of scores of nominal images. The score $S(X)$ is calculated as follows:

$$\hat{S}(X) := \max_{e \in \mathcal{P}(X)} \min_{e' \in \mathcal{M} \setminus \mathcal{P}(X)} d(e, e') \quad (2)$$

Similarly, a segmentation map can be computed by realigning the patch anomaly scores to match the original input resolution by upscaling the scores using bi-linear interpolation. Specifically, the

anomaly score at the pixel level for a pixel at coordinates $(i, j)$ in the image, with embedding $e_{i,j}$ is computed using the following equation:

$$\hat{S}(X_{i,j}) := \min_{e' \in \mathcal{M} \backslash \mathcal{P}(X)} d(e_{i,j}, e'), \tag{3}$$

Experimentally, the distribution of $\hat{S}(X)$ closely matches the one from a validation set. We evaluate four thresholding strategies on this estimated empirical distribution: *max*, outliers from a boxplot (*whisker*), percentile estimation from parametric distribution at $95th$ percentile (*beta-prime*-95), and percentile estimation from empirical distribution at $95th$ percentile (*empirical*-95). Additional details can be found in Supplementary Material.

## 5    Experimental setting

Our experimental setup assumes the unavailability of a validation set, reflecting real-world challenges. Furthermore, many VAD methods assume that the training data contains only nominal images, but the presence of contaminated training data with anomalies can significantly reduce performance [51].

Given the difficulty of avoiding such contamination in specialized domains due to annotation challenges, our setup transitions from many-shot to few-shot and finally to zero-shot settings by gradually reducing the number of examples in the training set until it is completely removed.

To adhere to our setup constraints, we employed pre-trained models without fine-tuning that operate effectively in low-data regimes as baselines. Specifically, we propose *Enhanced-Patchcore* for few-shot and many-shot settings. For the zero-shot setting, we use conversational VLMs including LLaVA 1.5-7B/13B and BakLLaVA-7B, [28], CogVLM-17B and CogVLM2-19B [47], and a VLM tailored for VAD, WinCLIP [21]. The prompt used to get VLMs' predictions is "*Is there any anomaly or defect in the image. Please answer by Yes or No.*". For WinCLIP, we use "cable" as the object to fill the templates. For the many-shot and few-shot tasks, $N$ images were randomly sampled from the training sections within the k-fold cross-validation. For the zero-shot task, the training sections were entirely discarded. The test sections remain constant within the k-fold across all tasks.

To evaluate our models' performance, we consider threshold-independent metrics Area Under the Precision-Recall curve (AUPR) and Area Under the Receiver Operating Characteristic Curve (AUROC), and threshold-dependent metrics: precision, recall, false positive rate (FPR), false negative rate (FNR) and F1-score at the image level. To compute AUROC and AUPR for conversational VLMs, we adapt the VQAScore [27] to obtain anomaly scores. Specifically, VQAScore computes the probability of the output token "*Yes*" when prompting VLMs with the fixed template "*Does this figure show [caption]? Please answer yes or no.*". We use "*an anomalous or defective cable*" as "*[caption]*". For per-pixel evaluation we use AUPRO [7]. Additional implementation details are in Supplementary Material.

## 6    Results and discussion

Table 1 summarizes the overall performance of the baseline models and *Enhanced-PatchCore* on our *CableInspect-AD* dataset at image-level. First, we can observe that CogVLM-17B has the best F1 Score, whereas CogVLM2-19B has the lowest FPR. They both outperform WinCLIP, for which threshold-dependent metrics cannot be computed without a validation set. Overall, VLMs show high AUROC and AUPR, highlighting their potential as effective anomaly detectors. *Enhanced-PatchCore* has a better F1 score than all VLMs except CogVLM-17B. There are large variations across VLMs, indicating the need for careful selection. CogVLM2-19B's higher AUROC and AUPR but worse F1 score suggest suboptimal thresholding, underscoring the challenge of effective threshold control in zero-shot VLMs. *Enhanced-PatchCore*, even with limited nominal images, maintains competitiveness while offering the added advantage of pixel-level evaluation.

**Performance variability in same category objects**    Figure 4 compares the threshold-dependent metrics on the *CableInspect-AD* dataset for each of the three cables. While all models achieve relatively high mean F1-score values, their performance can significantly vary (Figure 4a) across folds and cables. These variations are particularly notable for cables 2 and 3, which, being older, contain

Table 1: Performance metrics at image-level. Mean and standard deviation are calculated across all cables after averaging over all folds. VLMs and WinCLIP are evaluated in a zero-shot setting, while *Enhanced-PatchCore* is evaluated in a 100-shot setting using the *beta-prime*-95 thresholding strategy. Thresholded-metrics are not reported for WinCLIP since it necessitates a validation set.

| Model | F1 Score ↑ | FPR ↓ | AUPR ↑ | AUROC ↑ |
|---|---|---|---|---|
| LLaVA 1.5-7B | $0.59 \pm 0.07$ | $0.32 \pm 0.19$ | $0.75 \pm 0.05$ | $0.68 \pm 0.04$ |
| LLaVA 1.5-13B | $0.69 \pm 0.02$ | $0.66 \pm 0.21$ | $0.74 \pm 0.04$ | $0.66 \pm 0.03$ |
| BakLLaVA-7B | $0.69 \pm 0.02$ | $0.53 \pm 0.19$ | $0.77 \pm 0.04$ | $0.71 \pm 0.03$ |
| CogVLM-17B | $\mathbf{0.77 \pm 0.02}$ | $0.34 \pm 0.21$ | $0.83 \pm 0.03$ | $0.79 \pm 0.04$ |
| CogVLM2-19B | $0.66 \pm 0.04$ | $\mathbf{0.04 \pm 0.01}$ | $\mathbf{0.91 \pm 0.02}$ | $\mathbf{0.86 \pm 0.03}$ |
| WinCLIP | - | - | $0.76 \pm 0.06$ | $0.70 \pm 0.04$ |
| *Enhanced-PatchCore* | $0.75 \pm 0.03$ | $0.55 \pm 0.19$ | $0.84 \pm 0.06$ | $0.78 \pm 0.05$ |

artifacts like scratches and discoloration from natural wear. These artifacts were not considered as anomalies by the experts, posing a greater challenge. This underscores the uniqueness of our dataset, where objects of the same category can have a significantly variable appearance. Additionally, the performance varies across the folds because the test sets of each fold can differ in terms of anomaly types and grades (see Figure 2). Consequently, folds containing a higher proportion of harder-to-detect anomalies (e.g., long scratches) compared to easier ones might show lower performance. Furthermore, our analysis suggests that VLMs are more robust compared to other methods, showing more consistent performance across different folds and cables.

***Enhanced-PatchCore* - thresholding without a validation set**    From Figure 4a, we observe that the model performs well despite thresholding on the training set. Specifically, the performances of *Enhanced-PatchCore* in the few and many-shot settings employing various thresholding strategies show that the mean F1-score improves in most cases as the number of training images increases. Among the thresholding strategies, *max*—which is the most sensitive to outliers in the memory bank—appears brittle, while *whisker*, *empirical*-95 and *beta-prime*-95 seem to be more robust across the cables. Additionally, if we examine the precision-recall and FPR-FNR trade-offs, using the *beta-prime*-95 strategy as an example (Figures 4b and 4c), we observe that, overall, for cables 1 and 2, an increase in recall is accompanied by a decrease in precision, usually at the expense of an increase in FPR, accompanied by a decrease in FNR (i.e., 1 - Recall), as the number of training images increases. Moreover, increasing the number of images in the training set does not seem beneficial, as it increases the risk of including outliers in the memory bank. On the other hand, reducing the number of instances might result in a less diverse training set compared to the distribution of real-world nominal images.

**Analysis of conversational VLMs**    Table 1 shows that the VLMs achieve promising results despite not using any training examples (zero-shot). Specifically, the CogVLM variants outperform the other baselines. In Figure 4, CogVLM-17B shows the highest mean F1-score with the lowest variance across folds, outperforming other baselines across all cables (Figure 4a), whereas CogVLM2-19B shows the lowest FPR. Despite these encouraging results, VLMs are challenged by many limitations. Notably, VLMs can exhibit limitations in instruction following [20], be prone to object hallucinations [55], generate factual errors about objects, attributes, and relations [29], and be vulnerable to deceptive prompts [34]. Moreover, while conversational VLMs show promise in anomaly detection, their ability to accurately localize anomalies remains a challenge. To highlight some of these limitations, we present examples in Supplementary Material.

**Evaluating the impact of background removal**    One possible reason for the high variability of the performances of *Enhanced-PatchCore* is its sensitivity to variations in the background. Therefore, we evaluate the baseline models on a cropped version of *CableInspect-AD*, namely *CableInspect-AD_cropped*, in which we retain only the central part of the cables. In Figure 5, *Enhanced-PatchCore* shows lower variance in the F1-score across the different thresholding strategies while maintaining good performances on all cables. All thresholding strategies perform similarly, except for the *max*

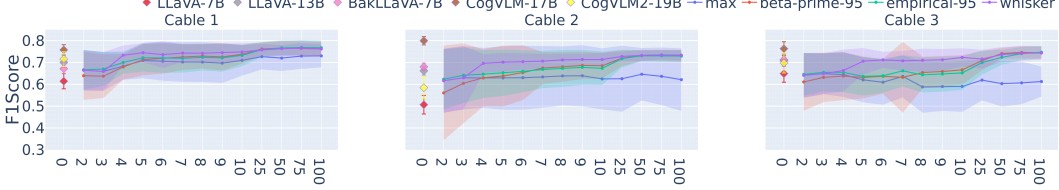

(a) F1-Score as the number of train images increases on the x-axis.

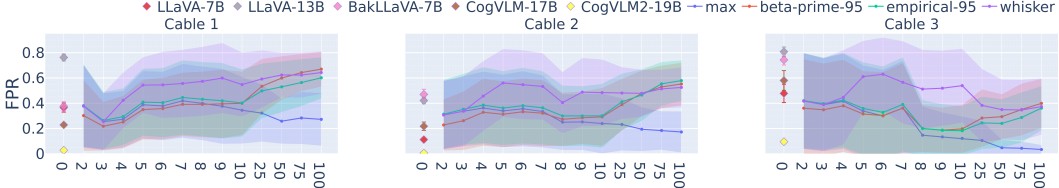

(b) FPR as the number of train images increases on the x-axis.

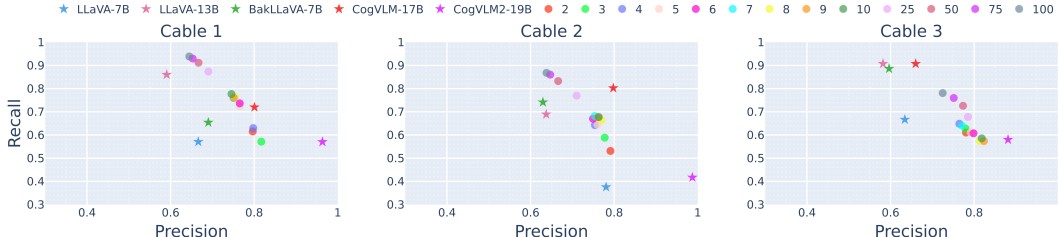

(c) Precision vs Recall. (*) show VLMs in a zero-shot setting. (o) show *Enhanced-PatchCore* with *beta-prime-95* thresholding. Here, the colors represent the number of training images.

Figure 4: Image-level results of *Enhanced-PatchCore* (few-/many-shot) with the thresholding strategies and conversational VLMs (zero-shot). (a) and (b) show the mean and standard deviation over all folds for F1-score and FPR for the three cables. The x-axis indicates the number of images in the training set. (c) shows mean precision vs mean recall over all folds.

strategy on cable 3. Thus, the extraction of the region of interest seems beneficial. Surprisingly, the performance of the conversational VLMs drop significantly. This could be attributed to the reduced view in the cropped version of the image, potentially making it more challenging for them.

In Figure 5b, we observe an increase in mean AUROC and a decrease in its variance as the number of training images increases, indicating that the choice of the training image in the few-shot setting can greatly influence the performance. However, the AUROC variance does not decrease when the background is retained. WinCLIP demonstrates enhancements in AUROC when excluding the background. Similar findings apply to AUPR. More details on metrics and visualizations are in Supplementary Material.

**Visual anomaly detection across different anomaly types and grades**    Despite the promising performances demonstrated by the baseline models, all the models fail to detect all types/grades of anomalies. For instance, Figure 6 shows the recall of anomalies based on type and grade by CogVLM-17B on the whole *CableInspect-AD* dataset. More pronounced anomaly types and grades such as *bent strand (important)* and *broken strand (complete)* are readily detected, whereas light and smaller anomalies such as *spaced strands* and *long scratches (light)* are prone to be overlooked. This highlights the importance of including multi-grade anomalies in the evaluation benchmark.

**Anomaly Segmentation**    *Enhanced-Patchcore* outperforms WinCLIP in the segmentation task on *CableInspect-AD_cropped*, with an AUPRO of $0.53 \pm 0.08$ compared to $0.27 \pm 0.06$ for WinCLIP. We apply thresholding strategies on anomaly maps generated by Enhanced-Patchcore to generate pixel-level predictions. We use a *max* thresholding strategy for the segmentation results shown in Figure 7 (more details are in the Supplementary material). The corresponding pixel-level metric, the

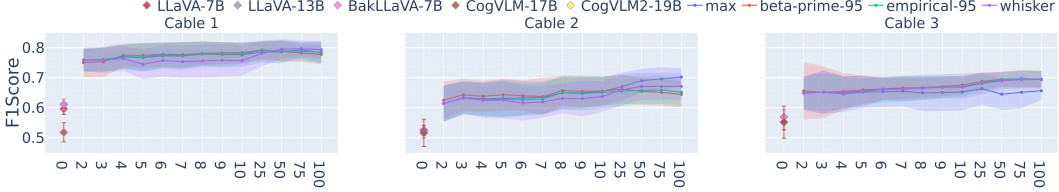

(a) F1-Score as the number of train images increases on the x-axis.

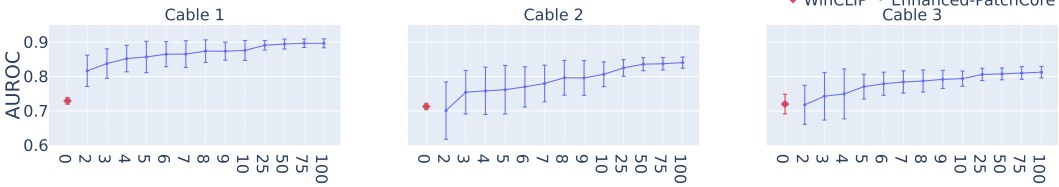

(b) AUROC as the number of train images increases on the x-axis for *Enhanced-PatchCore*.

Figure 5: Image-level results in zero-shot setting using conversational VLMs and WinCLIP, and, few-shot and many-shot using *Enhanced-PatchCore* on *CableInspect-AD_cropped*. Mean and standard deviation over all folds are reported for the three cables. On the figures, the x-axis indicates the number of images in the training set. (a) shows F1-score. For *Enhanced-PatchCore*, the metrics are computed using different thresholding strategies. (b) AUROC for *Enhanced-PatchCore* and WinCLIP.

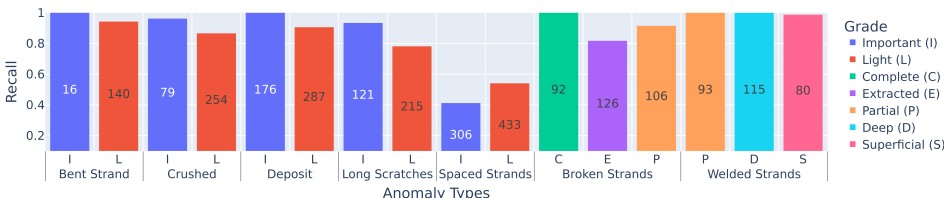

Figure 6: CogVLM-17B image-level recall per anomaly types/grades (sample counts on bars).

Pixel-wise Overlap (PRO) score, averaged across all cables and folds, is 0.28 ± 0.09. Figure 7 displays example outputs from *Enhanced-Patchcore*, illustrating that the model effectively identifies larger anomalies but struggles with subtler ones. The rightmost image shows a nominal image where texture changes from wear are visible. These texture variations can distract the model adding complexity to the task.

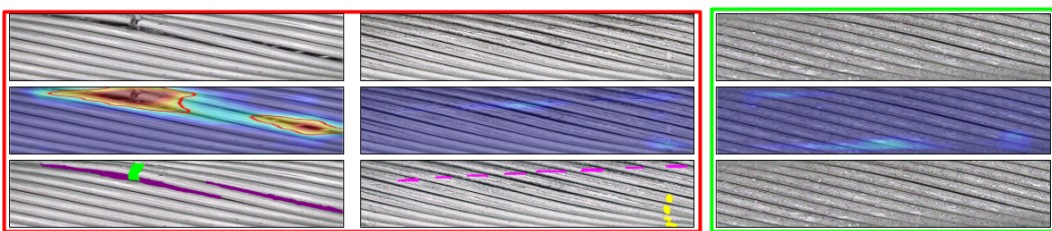

Figure 7: *Enhanced-PatchCore* qualitative results for anomaly segmentation. The rightmost image is nominal (green); the rest show anomalies (red). The images (top row) and pixel-level prediction heatmaps with contours of detected anomalies using the *max* thresholding strategy (middle row) are shown against ground truth masks (bottom row) from different cables. The bottom row shows the segmentation masks coloured based on the anomaly type. Some anomalies are easily detected (left column) whereas the others are difficult and are missed (middle column).

**Contribution**    Our dataset demonstrates its unique strength through the comprehensive diversity of anomaly types and severity levels it captures. Specifically, it includes seven distinct types of anomalies, each with up to three levels of severity. This allows for a more in-depth evaluation within the targeted domain. Broader datasets, with lower anomaly diversity per category, may not fully capture the intricacies persistent in real-world applications. In addition, given the accelerating electrification of transportation, there is a growing need for reliable transmission facilities. Therefore, it is critical to develop VAD models that can specialize in such high-stakes applications. Our dataset meets this need by offering a focused evaluation framework that complements broader datasets.

**Broad impact**    The methodologies and insights derived from our focused study are adaptable to a wide range of anomaly detection scenarios. For instance, our experiments demonstrate that Vision-Language Models (VLMs) can be effectively utilized for zero-shot VAD tasks. However, we also find that no current model performs well across all anomaly types, particularly when detecting light-grade anomalies. This finding reveals the limitations of current models and provides a valuable direction for future research aimed at enhancing model performance in specialized applications.

**Limitations**    We acknowledge that this work has the following limitations. First, we aimed to create a dataset containing a comprehensive range of real-world anomalies. However, this resulted in a higher anomaly ratio than typically observed in real-world scenarios, where anomalies rarely occur. This can be addressed by analyzing the results with this variation in mind or, when necessary, by employing stratified sampling to adjust the anomaly ratio within the folds. Second, despite our efforts to provide a rich and diverse set of examples for effective model learning and evaluation, the dataset does not encompass every possible anomaly found on a cable in real-world settings, because the methodology for data creation may not fully capture all complexities encountered in real-world scenarios, such as the deposition of snow or bird droppings on the cable.

**Ethical concerns**    We do not anticipate significant risks of security threats or human rights violations in our work or its potential applications. However, while our work aims to improve system reliability, we remind researchers that deploying machine learning models for VAD in robotic power line inspection may miss anomalies, potentially compromising safety and public utility operations.

# 7    Conclusion

In this work, we introduce *CableInspect-AD*, a novel anomaly detection dataset created and annotated by domain experts. We employ a k-fold evaluation to assess *Enhanced-PatchCore* with multiple thresholding strategies, WinCLIP and open VLMs on the proposed dataset. We find that, in general, the baselines show promising results in detecting anomalies on the cables, but struggle to detect anomalies of certain types and grades. This presents an important challenge for the development of new models on this task and highlights the potential value of *CableInspect-AD* as a resource for the broader AD community. Furthermore, we highlight the potential of recent open VLMs in zero-shot anomaly detection, requiring minimal prompt engineering and no image preprocessing. Future work will aim to assess VLM's zero-shot capabilities to other anomaly tasks such as type/grade classification, localization, and segmentation.

## Acknowledgments and Disclosure of Funding

This research was enabled in part by compute resources, software and technical help provided by Mila (mila.quebec). We thank Ali Harakeh and Pierre-Luc St-Charles from the Mila Applied Machine Learning Research Team (AMLRT) for fruitful discussions, brainstorming and feedback. We also thank Hydro-Québec and IREQ for their involvement throughout the project. The project received funding from Hydro-Québec and was further supported by governmental contributions from the Ministère de l'Économie, de l'Innovation et de l'Énergie (MEIE) and Innovation, Science and Economic Development Canada (ISED).

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
