## Supplementary Material for
## *CableInspect-AD: An Expert-Annotated Anomaly Detection Dataset*

We provide links to the dataset and the code repository for reproducibility in subsection A along with the author statement B. The detailed dataset documentation and intended uses in the form of a datasheet for datasets [17] are available in subsection J. We also include an ML reproducibility checklist (see S19).

In the following subsections, we present the dataset creation and annotation process (see C), the dataset partitioning using k-fold cross-validation (see D), a description of the thresholding strategies used (see E), more details on the background removal procedure (see F), implementation details (see G), threshold-independent metrics on *Enhanced-PatchCore* (see H), and a few qualitative examples obtained with VLMs (see I).

### A   Dataset and code access links

The project website link associated with the paper is the following: `https://mila-iqia.github.io/cableinspect-ad/`.

- **Dataset**: The dataset can be accessed via the *Data* icon/hyperlink in the project website: `https://mila-iqia.github.io/cableinspect-ad/`. The dataset is hosted and maintained by the authors. For more information, please refer to the *Distribution* and *Maintenance* subsections of the datasheet provided in J. The annotations are in the COCO format. We provide detailed explanations on how the dataset can be read in the code repository.

- **Code**: The link to the code repository is the following: `https://github.com/mila-iqia/cableinspect-ad-code`. The repository includes the code necessary to process the dataset, as well as the code required to reproduce all the experiments presented in the paper.

### B   Author statement

We, the authors of the submitted paper titled *CableInspect-AD: An Expert-Annotated Anomaly Detection Dataset*, hereby affirm the following:

- **Responsibility for Content**: We bear full responsibility for the content of this paper, including any potential violation of rights or legal issues arising from the use or distribution of the dataset described in our submission.

- **Data License Confirmation**: The dataset developed is licensed under Attribution NonCommercial ShareAlike 4.0 International License (CC BY-NC-SA 4.0).

### C   Dataset creation and annotation

Table S1 presents the anomalies annotation guidelines and Figure S1 presents the image acquisition process. The dataset underwent five iterative rounds of review and feedback, allowing the experts to reach a consensus. This process ensured that the final version was both reliable and reflective of real-world conditions. While very light anomalies, such as light deposits and scratches, might have been missed, the experts agreed these are not critical, as they would not require immediate repair in a real-world scenario and might even go undetected by experts. All mild and severe cases were thoroughly annotated. We did not quantify the annotation process' performance, as it was conducted in a consensus-driven, iterative manner until an agreement was reached.

### D   Dataset partitioning using k-fold cross-validation

The power line cable dataset is split into train and test sets using a k-fold sampling strategy based on defect identifiers. We consider each cable side independently, as anomalies with the same identifier do not often occur on both sides of the cable. Moreover, when an anomaly appears on both sides, its visual characteristics differ depending on the point of view.

To generate a fold, we start by randomly selecting a defect identifier and retrieving its corresponding images on the same cable side. This marks the beginning of the training section. In total, 100 nominal

Table S1: Anomaly types and grades annotation guidelines.

| Anomaly Type | Grade | Description |
|---|---|---|
| Welded strand | Superficial | Each strand is identifiable. |
| | Partial | Strands are fused together. |
| | Deep | A strand is completely disconnected by the fusion. |
| Broken strand | Partial | The strand is modified but still connected. |
| | Complete | The strand is completely cut but still in place. |
| | Extracted | The strand is cut; part of it is seen outside of the cable. |
| Spaced strand | Light | Slightly spaced, the next layer of strands cannot be seen. |
| | Important | Next layer of strands can be seen. |
| Bent strand | Light | Distortion smaller than the width of a strand. |
| | Important | Distortion bigger than the width of a strand. |
| Crushed | Light | Crushed part is smaller than the width of a strand. |
| | Important | Crushed part is bigger than the width of a strand. |
| Long scratch | Light | Scratch width smaller than 1/3 of a strand width. |
| | Important | Scratch width bigger than 1/3 of a strand width. |
| Deposit | Light | Deposit is smaller than the width of a strand. |
| | Important | Deposit is bigger than the width of a strand. |

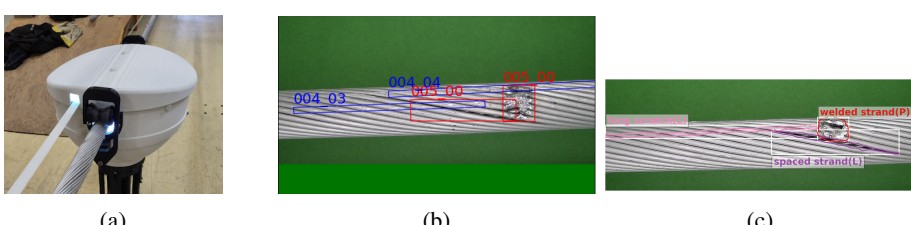

(a)       (b)       (c)

Figure S1: Image acquisition process. Image (a) shows the prototype of the apparatus used to control the background and the lighting during the acquisition phase. Image (b) shows an example of *CableInspect-AD* after post-processing and annotation. As we can see, a green band is added at the bottom of the image to cover the tape used for marking the location of the different anomalies, which was used during the annotation process. This measure aims to prevent the model from exploiting this information. The image has defects with more than one anomaly type. The defect labeled as 005_00 has multiple anomaly types: the left side of the defect is a light-spaced strand, while the right side is a partially welded strand. Furthermore, within this image, two additional defects can be identified: 004_03 and 004_04, both of which are light long scratches. Image (c) shows an example of pixel-level annotation.

images following this defect are included in the training set. We included 100 images to have a small training set in the same order of magnitude as the popular MVTec AD benchmark. The next defect identifier (following these 100 nominal images) marks the end of the training section. Images between the 100th image and the next defect identifier are discarded. To remove any overlap between cable sections in training and test sets, we use buffers before and after the training section (see Figure S2).

This process is repeated $k$ times, sequentially selecting defect identifiers from an ordered list spanning the entire cable length for each cable. Although each fold contains a constant number of training images (100), the number of test images and the anomaly ratios vary across folds.

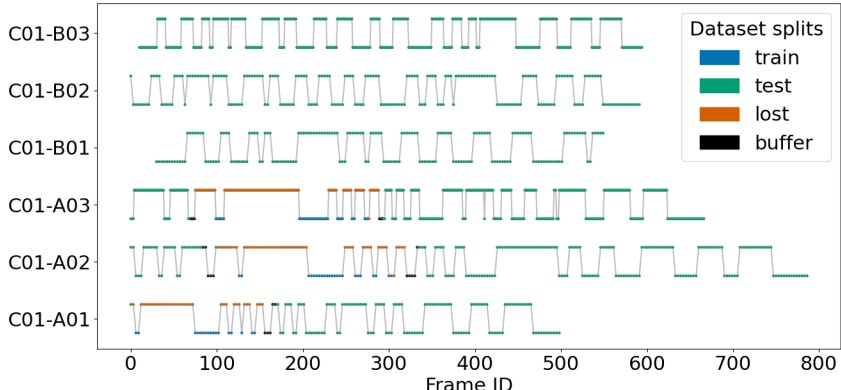

Figure S2: Example of one split in the k-fold for cable 1 (C01). Cable 1 has two sides, A and B, and three passes are done to capture the images 01, 02, and 03. Variations in the number of frames across cables result from slight fluctuations in the apparatus speed during manual acquisition and the fact that the cables are not of the same length. Additionally, initial frames showing poor quality were excluded from the dataset. Here, the lines represent the cable videos, and each dot within the lines represents a frame. The nominal images are at the lower level, while anomalous images are at the upper level (peaks) of the lines. Only nominal images are in the training set. Images in the training section that are not part of the training set are labeled as *lost*. Additionally, images associated with the two buffers are excluded. All remaining images, including those on the opposite side of the cable, constitute the test set. These images include both nominal and anomalous images.

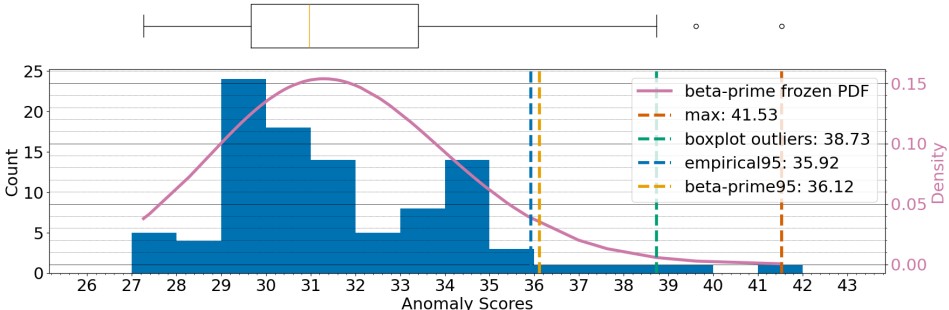

Figure S3: Thresholding strategies. The histogram shows an example of the empirical distribution of anomaly scores within the training set in a fold, with markers indicating the thresholds of the four thresholding strategies. Positioned above the histogram is the boxplot, which is used to set the *whisker* threshold. Additionally, superimposed on the histogram is the beta-prime fit of the training anomaly score distribution, used to set the *beta-prime* threshold. $\alpha = 95$ percentile is used to set both the *beta-prime* and *empirical* thresholds. Here, PDF stands for Probability Density Function.

## E  Thresholding strategies

To generate a threshold for the threshold-dependent metrics, we experiment with four thresholding strategies (see Figure S3):

- *Max*: The maximum anomaly score of the empirical distribution obtained from the training data is chosen as the threshold. Given the assumption that the training data contains only nominal images, this threshold should be lower than the scores associated with anomalies in the test set. However, in practice, the *max* strategy is sensitive to outliers in the training data.

- **Outliers from a boxplot (*whisker*)**: In a box-and-whisker plot, the points beyond the whiskers are considered outliers. To detect anomalies, the point at the upper quartile whisker is selected as the threshold. This corresponds to the largest anomaly score that is within $1.5 \times IQR$ above the third quartile ($Q3$), where $IQR$ is the interquartile range $Q3 - Q1$.

- **Percentile estimation from empirical distribution (*empirical-$\alpha$*)**: The observed anomaly scores are sorted, and the value corresponding to the $\alpha$ percentile is chosen as the threshold.
- **Percentile estimation from parametric distribution (*beta-prime-$\alpha$*)**: A beta-prime distribution is fit to the anomaly scores, and the value at $\alpha$ percentile is chosen as the threshold. By using a prior on the distribution family of the score, we expect the algorithm to be more robust in the low-data regime.

## F  Background removal

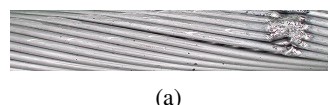

(a)

Figure S4: The image shows an example of *CableInspect-AD_cropped*.

We create *CableInspect-AD_cropped* dataset, containing the images with the background removed, keeping only the central part of the cables. The dataset was generated by extracting a central band of size $224 \times 1120$ as shown in fig. S4. During the ROI extraction, 696 anomalous images (typically the ones where anomalies extend outside the cable) out of the original 2639 become nominal, resulting in a dataset containing 4798 images: 2855 nominal and 1943 anomalous. Specifically, we lose ten unique anomalies, corresponding to six *broken strands (extracted)*, two *bent strands (light)*, one *broken strand (complete)*, and one *spaced strand (light)*. Furthermore, all the remaining 183 anomalies lose some of their views.

## G  Implementation details

Table S2: Main characteristics of the Vision-Language Models (VLMs) used in this work. The table provides details on each model, including the vision encoder (with its corresponding input image resolution in pixels), the Large Language Model (LLM) backbone, the multimodal alignment strategy, and the name of the weights used for inference from the *transformers* library (i.e., *HuggingFace* platform). For CogVLM variants, the *Visual Expert Module* refers to the vision-specific layers incorporated inside the LLM architecture to enhance multimodal alignment via deep fusion.

| Model | Vision Encoder | LLM Backbone | Multimodal Alignment | *HuggingFace* Weights |
|---|---|---|---|---|
| LLaVA-1.5-7B | CLIP-ViT-L/14 ($336^2$) | Vicuna-1.5-7B | MLP Projector | *llava-hf/llava-1.5-7b-hf* |
| LLaVA-1.5-13B | CLIP-ViT-L/14 ($336^2$) | Vicuna-1.5-13B | MLP Projector | *llava-hf/llava-1.5-13b-hf* |
| BakLLaVA-7B | CLIP-ViT-L/14 ($336^2$) | Mistral-7B | MLP Projector | *llava-hf/bakLlava-v1-hf* |
| CogVLM-17B | EVA02-CLIP-E ($490^2$) | Vicuna-1.5-7B | MLP Projector and a Visual Expert Module | *THUDM/cogvlm-chat-hf* |
| CogVLM2-19B | EVA02-CLIP-E ($1344^2$) | LLaMA-3-8B-Instruct | MLP Projector and a Visual Expert Module | *THUDM/cogvlm2-llama3-chat-19B* |

*Enhanced-PatchCore* was developed on top of PatchCore from *anomalib*[3] implementation with default hyperparameters. We sampled $n = 2, 3, 4, 5, 6, 7, 8, 9, 10$ images for few-shot and $n = 25, 50, 75, 100$ images for many-shot experiments as part of the training set, excluding the zero-shot scenario as it requires at least two images to constitute a memory bank in our enhanced version. We applied individual models for each cable to account for their distinct characteristics.

For VLMs we used the implementations from the *transformers*[4] library. LLaVA 1.5 is a recent iteration of the original LLaVA, with improvements in multimodal alignment by replacing the original linear projector with a two-layer MLP projector, as well as integrating academic task-oriented data into its training pipeline. BakLLaVA uses the same architecture as LLaVA 1.5, but replaces the

---
[3] https://github.com/openvinotoolkit/anomalib
[4] https://github.com/huggingface/transformers

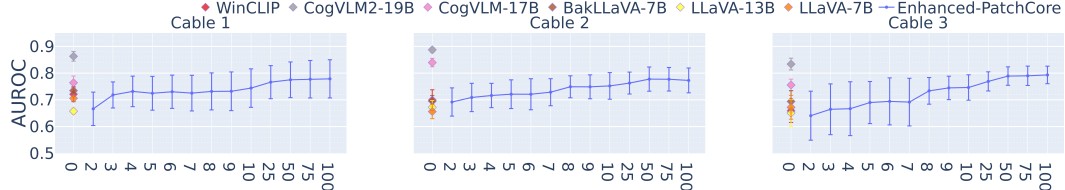

(a) Mean (+/- standard deviation) AUROC over all folds vs number of train images in x-axis.

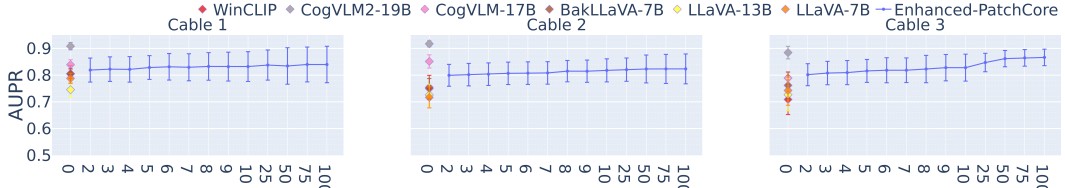

(b) Mean (+/- standard deviation) AUPR over all folds vs number of train images in x-axis.

Figure S5: The baseline VLMs and WinCLIP in zero-shot and *Enhanced-PatchCore* in few/many-shot setting results on *CableInspect-AD*. (a) and (b) show mean (+/- standard deviation) AUROC and AUPR over all folds for the three cables. The x-axis shows the number of images in the train set.

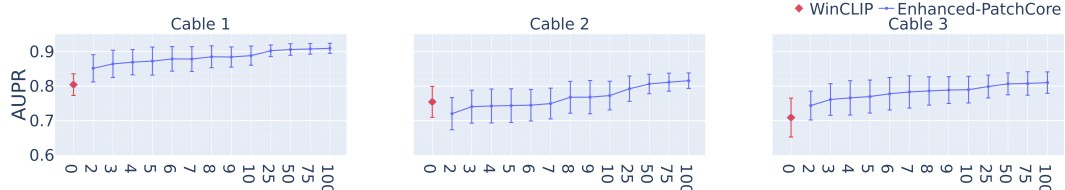

Figure S6: WinCLIP zero-shot and *Enhanced-PatchCore* in few/many-shot setting results on *CableInspect-AD_cropped*. The figures show the mean (+/- standard deviation) AUPR over all folds for the three cables. The x-axis shows the number of images in the train set.

Vicuna Large Language Model (LLM) backbone with a Mistral backbone instead. CogVLM models integrate a visual expert module inside the LLM backbone, enabling deeper fusion between the vision and language embeddings. Furthermore, they take higher input resolutions in their vision encoders and leverage a much larger pre-trained vision encoder. Table S2 highlights these differences.

For all VLMs model architectures tested, the inference was performed independently over all data samples. To prevent overfitting on the *CableInspect-AD* dataset in the zero-shot setting, we only conducted small-scale preliminary prompt optimization experiments on the MVTec AD dataset. Our experiments (results not shown) suggest that short and simple prompt instructions yield better results with the open VLMs.

The official implementation of WinCLIP is unavailable. We therefore use the implementation from *anomalib* and another implementation [5].

*Enhanced-PatchCore*, LLaVA 1.5, and BakLLaVA training and/or inference were performed on single-node NVIDIA GPUs (models A100, V100, and/or RTX8000), while CogVLM variants, inference was performed on single-node A100 GPUs (80GB). For VLMs inference, we make use of 4-bit quantization to reduce memory usage.

## H   Threshold-independent metrics on *Enhanced-PatchCore*

Figure S5 shows the performance of *Enhanced-PatchCore*, VLMs and WinCLIP on *CableInspect-AD* using two threshold-independent metrics: Area Under the ROC curve (AUROC) and Area Under the Precision-Recall curve (AUPR). CogVLM-19B outperforms all the baseline models. However, the

---
[5] https://github.com/caoyunkang/WinClip/blob/master/README.md

performance of the VLMs varies significantly. For *Enhanced-PatchCore*, we see an increase in the performance i.e., the mean metric increases as the number of training images increases. However, the variance does not decrease for all cables. For example, for cables 1 and 2, the variance of AUPR increases as the number of images increases in the train set. This could be due to the variations in the background. Contrastingly, the variance decreases when the background is removed in the images, as shown in Figure S6.

## I  Qualitative examples with VLMs

Figures S7 to S10 showcase capabilities and limitations of VLMs on the anomaly detection task. To complement the analysis, we also provide VLMs outputs for the generic image understanding task by prompting the models to describe the content of the image. We highlight three types of output information: (i) expected/correct information that aligns with the image's content or anomaly label, (ii) incorrect but plausible information (e.g. ambiguous), and (iii) incorrect information that does not align with the image's content or anomaly label (e.g. hallucinations).

For selected examples, we can observe that CogVLM-17B and CogVLM2-19B output more precise and/or refined descriptions of the cables and their anomalies (e.g. Figures S7 to S9). In contrast, we observe that LLaVA variants generate hallucinations more often than CogVLM variants (e.g. Figure S7), and show higher inconsistency between the anomaly detection task and image description task (e.g. Figures S8 and S9).

Similarly, Figures S11 to S18 show CogVLM-17B predictions for the anomaly detection task on the same sample ID taken from the two different dataset versions, *CableInspect-AD_raw* and *CableInspect-AD_cropped*. From the selected examples, we can observe instances where the VLM correctly identifies the sample as anomalous in both dataset versions (e.g. Figures S11, S13 to S15 and S17), and instances where the VLM correctly labels both samples as nominal (e.g. Figure S12). We can also observe that strand-related anomalies (e.g. spaced strands, broken strands) are more likely to be ignored in the cropped version. For instance, CogVLM-17B seems to be more sensitive to discoloration (e.g. welded strands accompanied by discoloration, see Figures S11 and S13 to S15) rather than pure structural changes (e.g. spaced strands, see Figure S18). Finally, nominal samples that present small defects not considered anomalous by the experts can lead to false positives in both versions of the dataset (e.g. Figure S16).

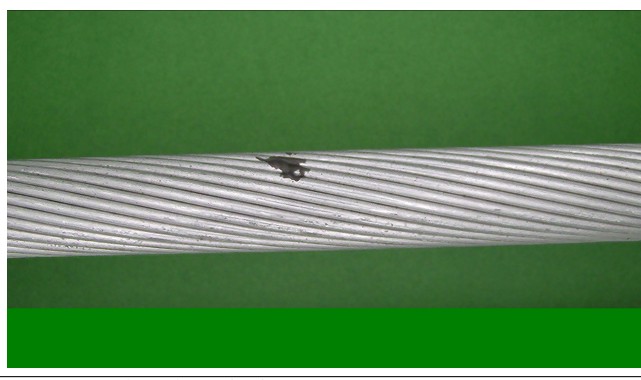

| | |
|---|---|
| **Label:** Anomalous. Type (grade): deposit (important). | |
| **Prompt:** "Is there an anomaly or defect in the image? Answer by either yes or no." | **Prompt:** "Describe the content of the image." |
| **LLaVA 1.5 7B:** Yes. | **LLaVA 1.5 7B:** The image features a close-up view of a silver wire, possibly a power cord or a cable. There is a small black spot on the wire, which appears to be a bird dropping. The bird is perched on the wire, possibly taking a break or resting. The scene captures the interaction between the bird and the wire, creating a unique and interesting visual. |
| **LLaVA 1.5 13B:** Yes. | **LLaVA 1.5 13B:** The image features a close-up view of a silver wire, possibly a speaker wire, with a black spot on it. The wire is positioned in the center of the image, and it appears to be made of metal. The wire is placed against a green background, which adds a contrasting color to the scene. |
| **BakLLaVA 7B:** Yes. | **BakLLaVA 7B:** The image features a close-up view of a metal wire, possibly a telephone or power line, against a green background. The wire appears to be slightly corroded, with some black spots visible on its surface. These black spots may indicate that the wire is old or has been exposed to certain elements, causing the corrosion. |
| **CogVLM 17B:** Yes. | **CogVLM 17B:** The image showcases a close-up view of a metallic object, possibly a wire or cable, with a distinctive grooved pattern. There's a small, dark spot or mark on the object. The background is a solid green color. |
| **CogVLM2 19B:** Yes. | **CogVLM2 19B:** The image showcases a close-up view of a cylindrical object with a ridged texture. The object appears to be made of metal, with a series of parallel grooves running along its length. There's a small, dark, irregularly shaped mark or stain on the surface of the object. The background is a solid green color. |

Figure S7: Example of VLM outputs (sample *C01_A01_frame_000108*). Colors highlight the correct, incorrect but plausible, and incorrect information related to the anomaly and/or cable description.

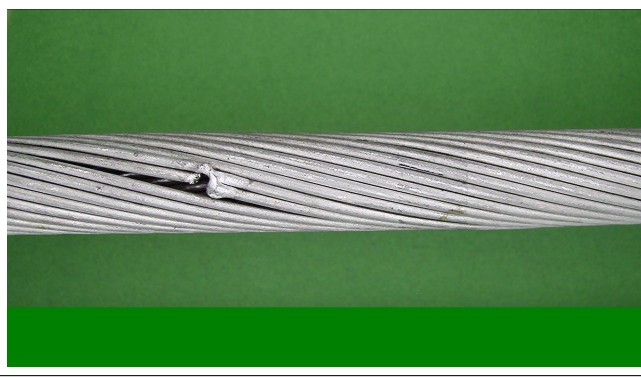

| Label: Anomalous. Type (grade): welded strands (deep), spaced strands (light). | |
|---|---|
| **Prompt:** "Is there an anomaly or defect in the image? Answer by either yes or no." | **Prompt:** "Describe the content of the image." |
| **LLaVA 1.5 7B:** Yes. | **LLaVA 1.5 7B:** The image features a close-up view of a silver wire, possibly a power cord or a cable. The wire is stretched out, and it appears to be in good condition. The green background provides a contrasting color to the silver wire, making it stand out. The focus of the image is on the wire, highlighting its texture and appearance. |
| **LLaVA 1.5 13B:** Yes. | **LLaVA 1.5 13B:** The image features a large, silver metal cable or wire, which appears to be a piece of rope or a thick wire. The cable is lying on a green background, possibly a table or a surface. The cable is quite long, extending from the left side of the image to the right side, covering most of the background. |
| **BakLLaVA 7B:** Yes. | **BakLLaVA 7B:** The image features a close-up view of a large, thick cable, possibly an electrical or telephone wire, against a green background. The cable appears to be slightly corroded, showing signs of wear and tear. |
| **CogVLM 17B:** Yes. | **CogVLM 17B:** The image showcases a close-up view of a coiled wire or cable. The wire appears to be made of a metallic material, possibly steel or aluminum, and is tightly wound in a helical pattern. There's a noticeable damage or break in one of the coils, with a small hole visible. |
| **CogVLM2 19B:** Yes. | **CogVLM2 19B:** The image showcases a close-up view of a cylindrical object, which appears to be made of tightly wound metal or wire. The object has a shiny, metallic surface with a series of parallel ridges. There's a noticeable section where the winding seems to have been interrupted, revealing a gap or break in the structure. |

Figure S8: Example of VLM outputs (sample *C03_A01_frame_000123*). Colors highlight the correct, incorrect but plausible, and incorrect information related to the anomaly and/or cable description.

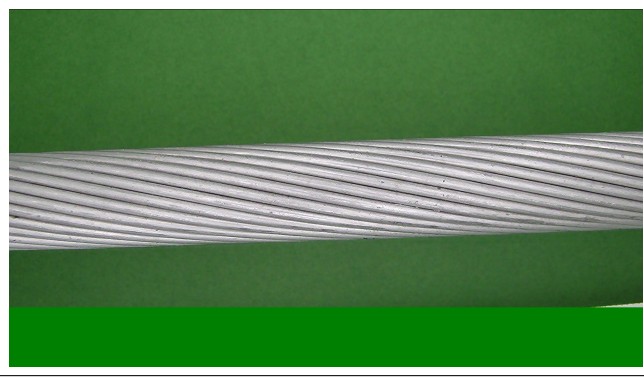

| **Label:** Nominal. | |
|---|---|
| **Prompt:** "Is there an anomaly or defect in the image? Answer by either yes or no." | **Prompt:** "Describe the content of the image." |
| **LLaVA 1.5 7B:** No. | **LLaVA 1.5 7B:** The image features a large, long, and thin wire, possibly a power cable or a telephone wire, stretched across the entire frame. The wire is silver in color and appears to be a part of a larger structure. The green background provides a contrasting and vibrant backdrop for the wire, making it stand out prominently. |
| **LLaVA 1.5 13B:** Yes. | **LLaVA 1.5 13B:** The image features a large spool of wire, which is made of metal and has a silver color. The spool is placed on a green background, which provides a contrasting backdrop. The wire is wound tightly around the spool, creating a neat and organized appearance. |
| **BakLLaVA 7B:** No. | **BakLLaVA 7B:** The image features a very long and flexible piece of white rope, possibly made of nylon or another synthetic material. The rope is in a close-up view, making it the main focus of the image. It is set against a green background, which provides a nice contrast to the white rope. |
| **CogVLM 17B:** No. | **CogVLM 17B:** The image showcases a close-up view of a cylindrical object with a repetitive pattern of ridges or grooves. The object appears to be made of a metallic material, possibly steel or aluminum, given its shiny and smooth surface. The background is a solid green color, which contrasts with the metallic hue of the object. |
| **CogVLM2 19B:** No. | **CogVLM2 19B:** The image showcases a close-up view of a cylindrical object with a ridged texture. The object appears to be made of a metallic material, with parallel ridges running along its length. The background is a solid green color. |

Figure S9: Example of VLM outputs (sample *C01_B02_frame_000303*). Colors highlight the correct, incorrect but plausible, and incorrect information related to the anomaly and/or cable description.

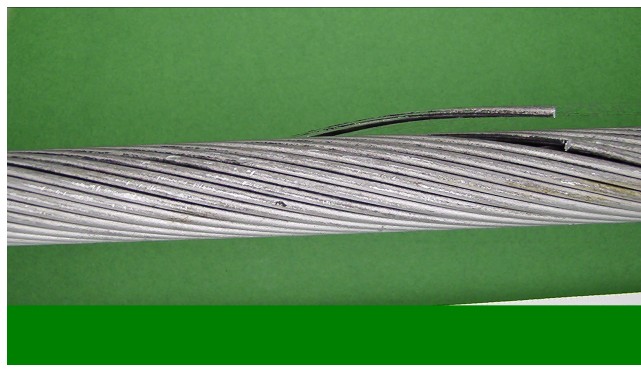

| **Label:** Anomalous. Type (grade): broken strands (complete), broken strands (extracted), spaced strands (important). | |
| --- | --- |
| **Prompt:** "Is there an anomaly or defect in the image? Answer by either yes or no." | **Prompt:** "Describe the content of the image." |
| **LLaVA 1.5 7B:** Yes. | **LLaVA 1.5 7B:** The image features a close-up view of a wire, possibly a power cord or a guitar string. The wire is stretched out, and it appears to be a black color. The wire is the main focus of the image, and it is displayed prominently in the foreground. |
| **LLaVA 1.5 13B:** Yes. | **LLaVA 1.5 13B:** The image features a close-up view of a coiled wire, possibly a speaker wire, lying on a green background. The wire is tightly coiled, and its color is a combination of black and silver. The wire's texture and pattern create an interesting visual effect, making it stand out against the green background. |
| **BakLLaVA 7B:** Yes. | **BakLLaVA 7B:** The image features a close-up view of a large, thick wire or cable, possibly made of metal, lying on a green background. The wire occupies a significant portion of the frame, with its length extending from the left to the right side of the image. |
| **CogVLM 17B:** Yes. | **CogVLM 17B:** The image showcases a close-up view of a metallic wire or cable with a distinctive pattern of concentric circles. The wire appears to be slightly frayed or damaged at one end. |
| **CogVLM2 19B:** Yes. | **CogVLM2 19B:** The image showcases a close-up view of a cylindrical object, possibly a metal rod or wire, with a series of tightly wound, parallel strands. There's a small, curved piece of the same material detached and placed beside the main object. The background is a solid green color. |

Figure S10: Example of VLM outputs (sample *C03_A02_frame_000604*). Colors highlight the correct, incorrect but plausible, and incorrect information related to the anomaly and/or cable description.

| *CableInspect-AD_raw* | *CableInspect-AD_cropped* |
|---|---|
| 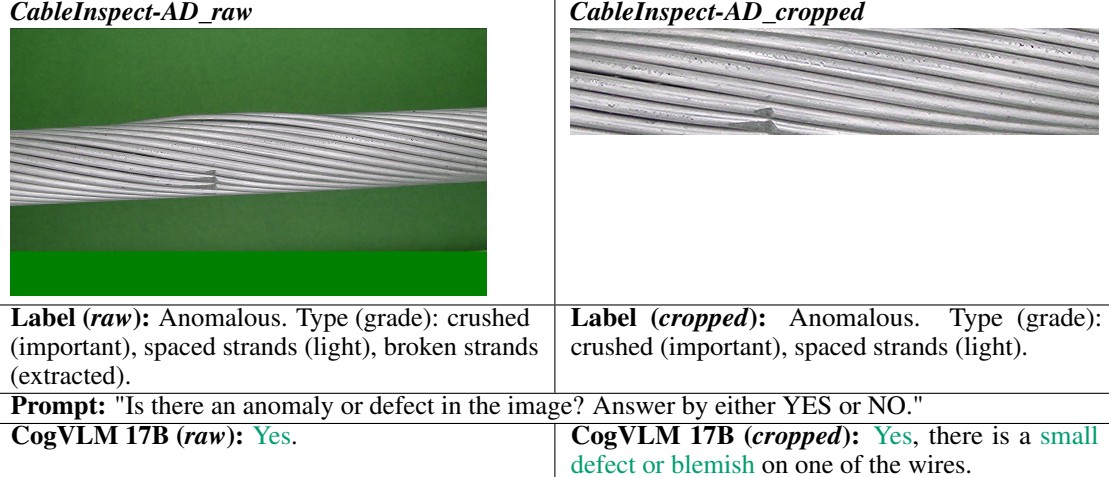 | |
| **Label (*raw*):** Anomalous. Type (grade): crushed (important), spaced strands (light), broken strands (extracted). | **Label (*cropped*):** Anomalous. Type (grade): crushed (important), spaced strands (light). |
| **Prompt:** "Is there an anomaly or defect in the image? Answer by either YES or NO." | |
| **CogVLM 17B (*raw*):** Yes. | **CogVLM 17B (*cropped*):** Yes, there is a small defect or blemish on one of the wires. |

Figure S11: Example of CogVLM-17B outputs (sample *C01_B01_frame_000213*) for the *raw* and *cropped* versions of the sample. In this example, both versions are correctly identified as anomalous. Colors highlight the correct, incorrect but plausible, and incorrect output information.

| *CableInspect-AD_raw* | *CableInspect-AD_cropped* |
|---|---|
| 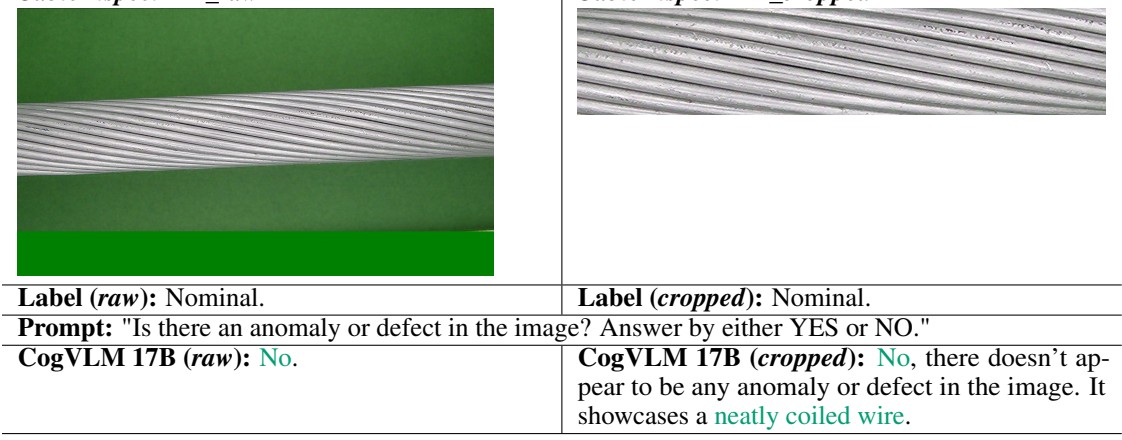 | |
| **Label (*raw*):** Nominal. | **Label (*cropped*):** Nominal. |
| **Prompt:** "Is there an anomaly or defect in the image? Answer by either YES or NO." | |
| **CogVLM 17B (*raw*):** No. | **CogVLM 17B (*cropped*):** No, there doesn't appear to be any anomaly or defect in the image. It showcases a neatly coiled wire. |

Figure S12: Example of CogVLM-17B outputs (sample *C01_B01_frame_000246*) for the *raw* and *cropped* versions of the sample. In this example, both versions are correctly identified as nominal. Colors highlight the correct, incorrect but plausible, and incorrect output information.

| *CableInspect-AD_raw* | *CableInspect-AD_cropped* |
|---|---|
| 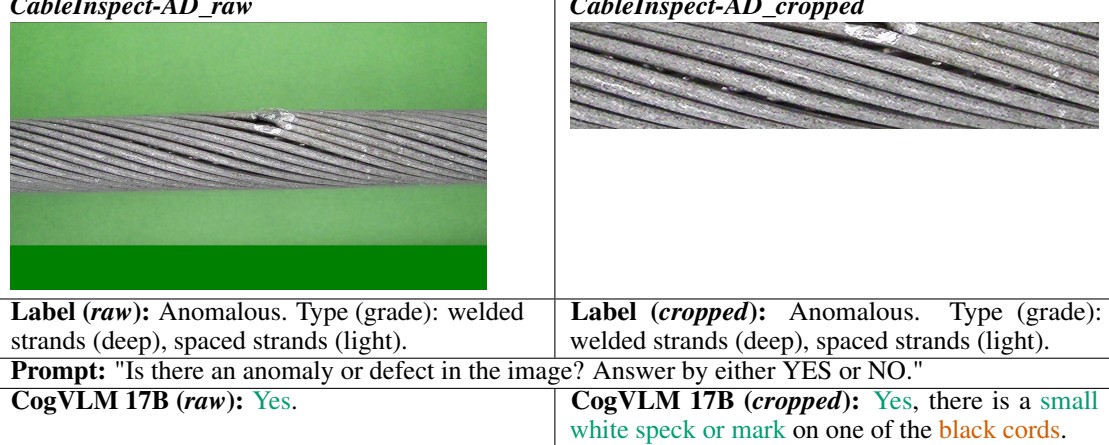 | |
| **Label (*raw*):** Anomalous. Type (grade): welded strands (deep), spaced strands (light). | **Label (*cropped*):** Anomalous. Type (grade): welded strands (deep), spaced strands (light). |
| **Prompt:** "Is there an anomaly or defect in the image? Answer by either YES or NO." | |
| **CogVLM 17B (*raw*):** Yes. | **CogVLM 17B (*cropped*):** Yes, there is a small white speck or mark on one of the black cords. |

Figure S13: Example of CogVLM-17B outputs (sample *C02_A01_frame_000257*) for the *raw* and *cropped* versions of the sample. In this example, both versions are correctly identified as anomalous. Colors highlight the correct, incorrect but plausible, and incorrect output information.

| *CableInspect-AD_raw* | *CableInspect-AD_cropped* |
|---|---|
| 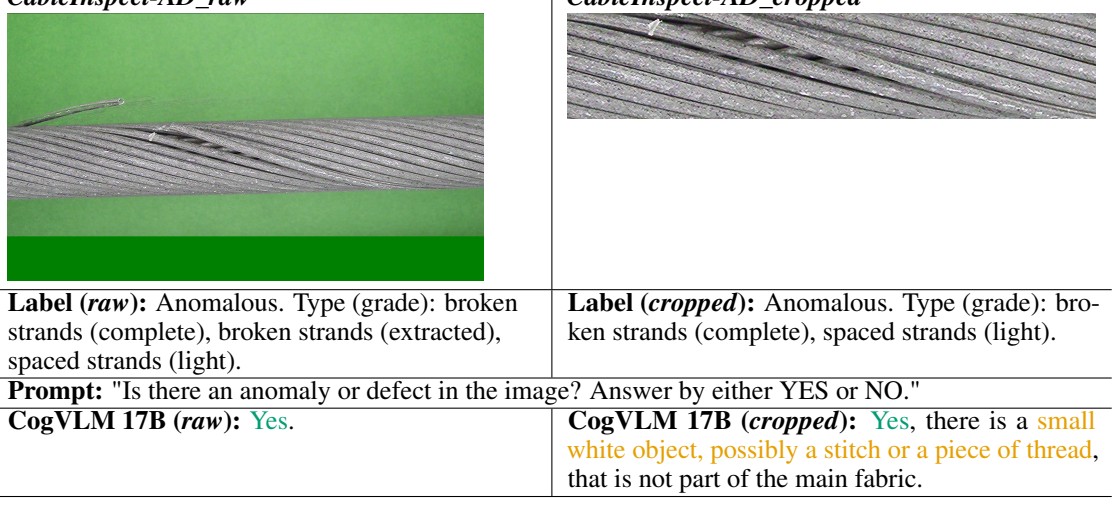 | |
| **Label (*raw*):** Anomalous. Type (grade): broken strands (complete), broken strands (extracted), spaced strands (light). | **Label (*cropped*):** Anomalous. Type (grade): broken strands (complete), spaced strands (light). |
| **Prompt:** "Is there an anomaly or defect in the image? Answer by either YES or NO." | |
| **CogVLM 17B (*raw*):** Yes. | **CogVLM 17B (*cropped*):** Yes, there is a small white object, possibly a stitch or a piece of thread, that is not part of the main fabric. |

Figure S14: Example of CogVLM-17B outputs (sample *C02_A01_frame_000578*) for the *raw* and *cropped* versions of the sample. In this example, the broken strands (extracted) anomaly is lost in the *cropped* version. Colors highlight the correct, incorrect but plausible, and incorrect output information.

| CableInspect-AD_raw | CableInspect-AD_cropped |
|---|---|

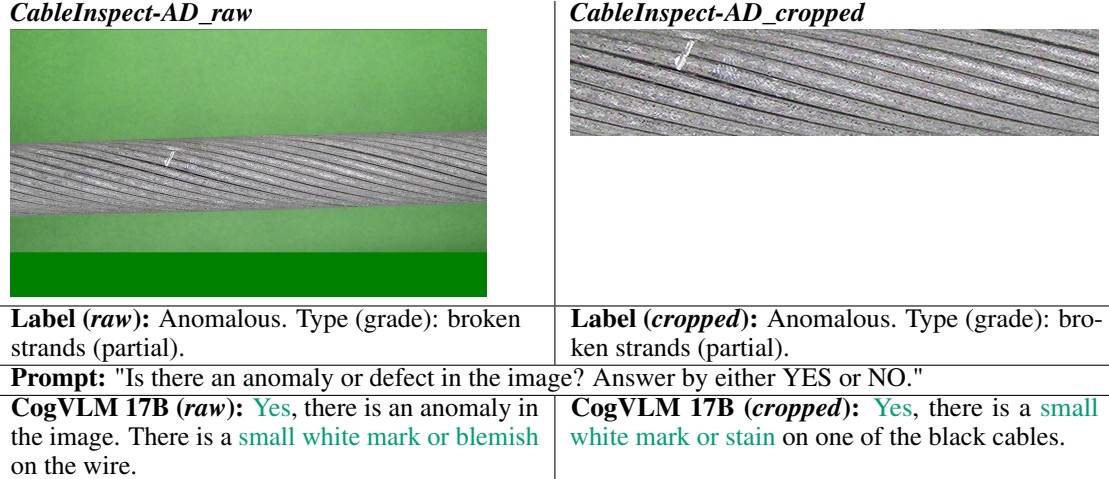

| **Label (*raw*):** Anomalous. Type (grade): broken strands (partial). | **Label (*cropped*):** Anomalous. Type (grade): broken strands (partial). |
|---|---|
| **Prompt:** "Is there an anomaly or defect in the image? Answer by either YES or NO." ||
| **CogVLM 17B (*raw*):** Yes, there is an anomaly in the image. There is a small white mark or blemish on the wire. | **CogVLM 17B (*cropped*):** Yes, there is a small white mark or stain on one of the black cables. |

Figure S15: Example of CogVLM-17B outputs (sample *C02_A01_frame_000635*) for the *raw* and *cropped* versions of the sample. In this example, both versions are correctly identified as anomalous. Colors highlight the correct, incorrect but plausible, and incorrect output information.

| CableInspect-AD_raw | CableInspect-AD_cropped |
|---|---|

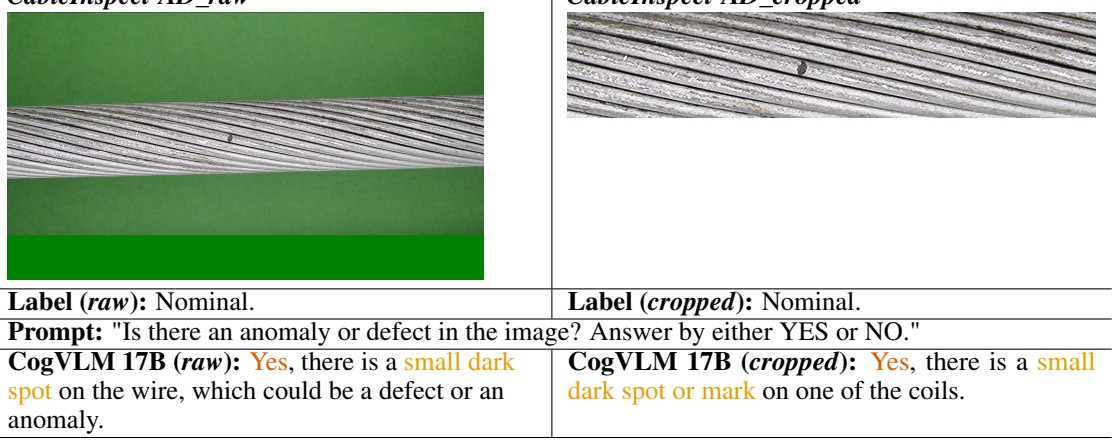

| **Label (*raw*):** Nominal. | **Label (*cropped*):** Nominal. |
|---|---|
| **Prompt:** "Is there an anomaly or defect in the image? Answer by either YES or NO." ||
| **CogVLM 17B (*raw*):** Yes, there is a small dark spot on the wire, which could be a defect or an anomaly. | **CogVLM 17B (*cropped*):** Yes, there is a small dark spot or mark on one of the coils. |

Figure S16: Example of CogVLM-17B outputs (sample *C03_A01_frame_000429*) for the *raw* and *cropped* versions of the sample. In this example, both versions are identified as anomalous even though the expert labeling is nominal. Colors highlight the correct, incorrect but plausible, and incorrect output information.

| *CableInspect-AD_raw* | *CableInspect-AD_cropped* |
|---|---|
| 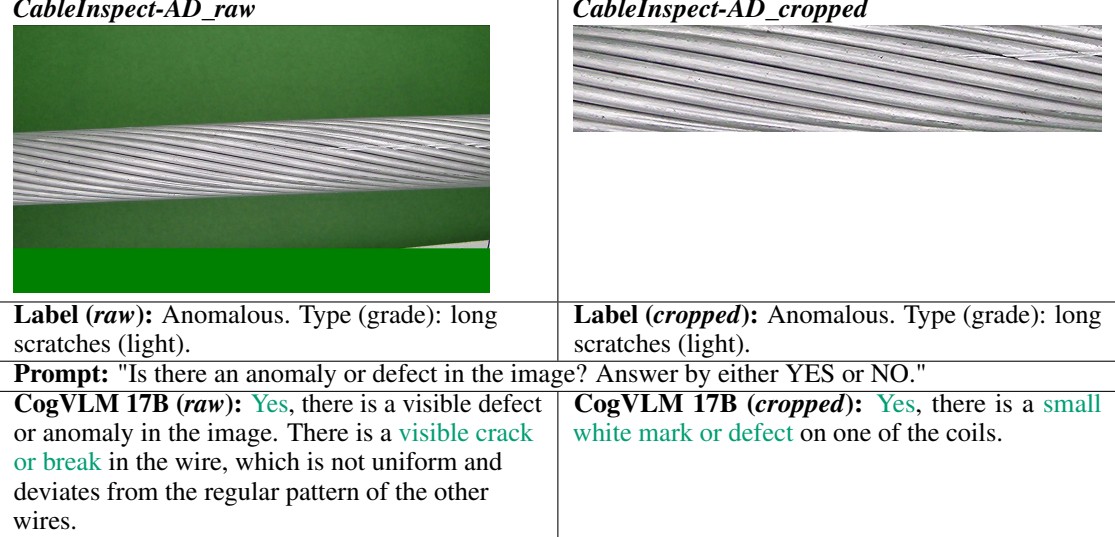 | |
| **Label (*raw*):** Anomalous. Type (grade): long scratches (light). | **Label (*cropped*):** Anomalous. Type (grade): long scratches (light). |
| **Prompt:** "Is there an anomaly or defect in the image? Answer by either YES or NO." ||
| **CogVLM 17B (*raw*):** Yes, there is a visible defect or anomaly in the image. There is a visible crack or break in the wire, which is not uniform and deviates from the regular pattern of the other wires. | **CogVLM 17B (*cropped*):** Yes, there is a small white mark or defect on one of the coils. |

Figure S17: Example of CogVLM-17B outputs (sample *C01_B02_frame_000195*) for the *raw* and *cropped* versions of the sample. In this example, the anomaly is correctly identified in both versions. Colors highlight the correct, incorrect but plausible, and incorrect output information.

| *CableInspect-AD_raw* | *CableInspect-AD_cropped* |
|---|---|
| 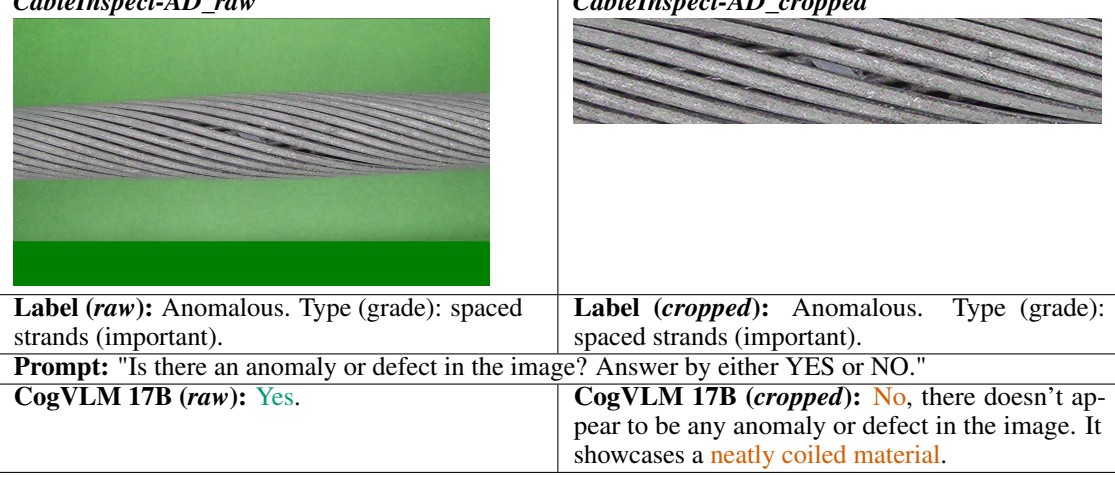 | |
| **Label (*raw*):** Anomalous. Type (grade): spaced strands (important). | **Label (*cropped*):** Anomalous. Type (grade): spaced strands (important). |
| **Prompt:** "Is there an anomaly or defect in the image? Answer by either YES or NO." ||
| **CogVLM 17B (*raw*):** Yes. | **CogVLM 17B (*cropped*):** No, there doesn't appear to be any anomaly or defect in the image. It showcases a neatly coiled material. |

Figure S18: Example of CogVLM-17B outputs (sample *C02_B02_frame_000746*) for the *raw* and *cropped* versions of the sample. In this example, the anomaly related to the spaced strands is not detected in the *cropped* version. Colors highlight the correct, incorrect but plausible, and incorrect output information.

## J    Datasheet

**Motivation**

### For what purpose was the dataset created?

The dataset was created to enable research on industrial anomaly detection for robotic transmission cable inspection.

### Who created this dataset and on behalf of which entity?

The dataset was created by the authors Jean-Luc Bedwani and Nicolas Pouliot on behalf of IREQ - Institut de recherche d'Hydro-Québec.

### Who funded the creation of the dataset?

The funding for the creation of the dataset was provided by Hydro-Québec and was further supported by governmental contributions from the Ministère de l'Économie, de l'Innovation et de l'Énergie (MEIE) and Innovation, Science and Economic Development Canada (ISED).

### Any other comments?

None.

**Composition**

### What do the instances that comprise the dataset represent?

The dataset comprises high-resolution (1920×1080 pixels) close-up RGBA images of three real power line cables with a focus on anomaly detection (both nominal and anomalous images are included). The anomalous images showcase seven types of anomalies with different grades comprising 110 manually created and 83 pre-existing real-world anomalies from various angles, providing a realistic setting for robotic inspection.

### How many instances are there in total?

The dataset contains 4,798 annotated images: 2,639 anomalous and 2,159 nominal images. Anomalous images include 193 unique anomalies, comprising 110 manually created and 83 pre-existing real-world anomalies.

### Does the dataset contain all possible instances or is it a sample of instances from a larger set?

The dataset consists of images extracted from three videos recorded at a frame rate of 30 frames per second. To facilitate anomaly annotations, one frame out of three was retained, resulting in a reduced frame rate of 10 frames per second.

### What data does each instance consist of?

Each instance consists of an image accompanied by labels indicating whether it is nominal or anomalous. Anomalous instances also include expert annotations, including bounding boxes, type, and grade annotations for each anomaly. Additionally, per-pixel labels for the first video of each cable are included.

### Is there a label or target associated with each instance?

Yes.

### Is any information missing from individual instances?

No.

**Are relationships between individual instances made explicit?**

Yes. The instances come from two sides (labeled A and B) of three cables (labeled C01, C02, C03). For each cable side, three videos were recorded (labeled 01, 02, 03) at a frame rate of 30 frames per second, with frames numbered sequentially. Instances are labeled as follows: *{cable number}_{cable side and video number}_frame_{frame number}* (e.g., *C02_B01_frame_000376*).

**Are there recommended data splits?**

Yes. We propose splitting the dataset into train and test sets using a k-fold cross-validation sampling strategy based on defect identifiers where each cable side is considered independently. We provide the split in the case where the training set includes 100 images.

**Are there any errors, sources of noise, or redundancies in the dataset?**

Yes. The dataset contains redundancies as each cable side has been recorded three times, resulting in multiple instances covering the same part of the cable with slight variations. Also, the slow frame rate causes consecutive frames to overlap. Moreover, the dataset can contain errors and noise in the annotations, particularly for light and smaller anomalies, which can be challenging for experts to detect and annotate. The labels can be noisy, as bounding boxes lack precision, and there may be mislabeling in anomaly type/grade.

**Is the dataset self-contained, or does it link to or otherwise rely on external resources?**

The dataset is self-contained and does not rely on external resources.

**Does the dataset contain data that might be considered confidential?**

No.

**Does the dataset contain data that, if viewed directly, might be offensive, insulting, threatening, or might otherwise cause anxiety?**

No.

**Any other comments?**  None.

---

| Collection Process |
|:---:|

**How was the data associated with each instance acquired?**

The data associated with each instance was acquired through a meticulous manual process. Experts identified seven types of anomalies from actual cables in operation, each categorized by severity grades. These anomalies were manually created by experts on three real power line cables, each referenced with a unique identifier, and assigned to the corresponding anomaly types. To optimize the cable usage, experts have utilized both sides of the cables (up and down), referred to as sides A and B, respectively. The cables are suspended for image acquisition, and a realistic apparatus is used to capture the images to ensure a uniform background. Along each cable, a tape with markers identifies the location of different anomalies to ease the annotation process.

**What mechanisms or procedures were used to collect the data?**

For each cable side, three videos were recorded, captured at a frame rate of 30 frames per second, composed of RGBA images of 1920×1080 pixels. In total, 18 videos were recorded by manually moving a camera along the cables at different speeds.

**If the dataset is a sample from a larger set, what was the sampling strategy?**

The dataset consists of images extracted from three videos recorded at a frame rate of 30 frames per second. To facilitate anomaly annotations, one frame out of three was retained, resulting in a reduced frame rate of 10 frames per second.

**Who was involved in the data collection process and how were they compensated?**

The authors Jean-Luc Bedwani and Nicolas Pouliot collected the data as part of their employment at their institution IREQ - Institut de recherche d'Hydro-Québec.

**Over what timeframe was the data collected?**

The dataset was collected on actual cables within a few days.

**Were any ethical review processes conducted?**

Not applicable.

**Any other comments?**

None.

---

**Preprocessing/cleaning/labeling**

---

**Was any preprocessing/cleaning/labeling of the data done?**

Yes.

- Along each cable, a tape with markers identifies the location of different anomalies to ease the annotation process. A green band is added to cover the tape during post-processing to prevent the model from exploiting this information.

- For each video, one frame out of three was retained, resulting in a reduced frame rate of 10 frames per second.

- Initial frames showing poor quality were excluded from the dataset.

- The labeling was done by the experts and consists of bounding boxes used to locate the anomalies. The anomaly type and grade are assigned based on the appearance of the anomaly in the image, which matches the description defined by the experts. An image containing at least one anomaly is considered anomalous. Pixel-level annotations are generated using SAM with expert annotated bounding boxes as inputs. This is followed by manual correction. Authors and other experts from IREQ - Institut de recherche d'Hydro-Québec were involved in the labeling process. More specifically, the dataset was annotated by at least four IREQ experts who first developed and agreed on guidelines to establish a clear annotation framework. The dataset then underwent five iterative rounds of review and feedback, allowing the experts to reach a consensus. This process ensured that the final version is both reliable and reflective of real-world conditions. While very light anomalies, such as light deposits and scratches, might have been missed, the experts agreed these are not critical, as they would not require immediate repair in a real-world scenario and might even go undetected by experts. All mild and severe cases were thoroughly annotated. We did not quantify the annotation process' performance, as it was conducted in a consensus-driven, iterative manner until an agreement was reached.

- Two versions of the labels are released: (1) the bounding boxes with expert annotations. (2) pixel-level annotations. The annotations are available in COCO format.

**Was the "raw" data saved in addition to the preprocessed/cleaned/labeled data?**

Yes. The "raw" data was saved. However, we only provide the resampled dataset as raw version. This resampled dataset has a frame rate of 10 frames per second, a green band that covers the tape, and excludes the low-quality frames.

**Is the software used to preprocess/clean/label the instances available?**

Yes. We used the Computer Vision Annotation Tool (CVAT) and Python scripts.

**Any other comments?**

None.



**Uses**



**Has the dataset been used for any tasks already?**

The dataset has been developed and used for the task of industrial anomaly detection and segmentation in the context of robotic power line cable inspection.

**Is there a repository that links to any or all papers or systems that use the dataset?**

Yes. Refer to the project website: `https://mila-iqia.github.io/cableinspect-ad/`.

**What (other) tasks could the dataset be used for?**

The dataset could potentially be used for research on other anomaly tasks such as type/grade classification, and localization since the annotations are compatible with these tasks as well.

**Is there anything about the composition of the dataset or the way it was collected and preprocessed/cleaned/labeled that might impact future uses?**

Yes. When building the dataset, we attempt to include a comprehensive range of real-world anomalies. However, this leads to a higher anomaly ratio than what is typically observed in real-world scenarios, where anomalies are rare. Additionally, despite our efforts to provide a rich set of diverse examples for effective model learning and evaluation, the dataset may not encompass every possible anomaly that may appear on a cable in a real-world setting.

**Are there tasks for which the dataset should not be used?**

None that we are aware of.

**Any other comments?** None.



**Distribution**



**Will the dataset be distributed to third parties outside of the entity on behalf of which the dataset was created?**

Yes. The dataset is publicly available on the internet through the project website: `https://mila-iqia.github.io/cableinspect-ad/`.

**How will the dataset be distributed?**

The dataset is accessible through the project website: `https://mila-iqia.github.io/cableinspect-ad/`.

**When will the dataset be distributed?**

The dataset is available and is accessible through the project website: `https://mila-iqia.github.io/cableinspect-ad/`.

**Will the dataset be distributed under a copyright or other intellectual property (IP) license, and/or under applicable terms of use (ToU)?**

Yes. We release *CableInspect-AD* in the public domain under CC BY-NC-SA 4.0 license. More details are on the project website.

**Have any third parties imposed IP-based or other restrictions on the data associated with the instances?**

None that we are aware of.

**Do any export controls or other regulatory restrictions apply to the dataset or to individual instances?**

None that we are aware of.

**Any other comments?**

None.

---

**Maintenance**

---

**Who will be supporting/hosting/maintaining the dataset?**

The authors will support and maintain the dataset.

**How can the owner/curator/manager of the dataset be contacted?**

Contact the authors.

**Is there an erratum?**

No. Future updates (if any) will be specified in the project website.

**Will the dataset be updated?**

Currently, no updates are planned.

**If the dataset relates to people, are there applicable limits on the retention of the data associated with the instances?**

Not applicable.

**Will older versions of the dataset continue to be supported/hosted/maintained?**

Yes. In the case of updates, refer to the project website: `https://mila-iqia.github.io/cableinspect-ad/`.

**If others want to extend/augment/build on/contribute to the dataset, is there a mechanism for them to do so?**

Yes. Suggestions for the augmentation of the dataset can be made by contacting the authors.

**Any other comments?**

None.


## The Machine Learning Reproducibility Checklist (v2.0, Apr.7 2020)

For all **models** and **algorithms** presented, check if you include:
- ☒ A clear description of the mathematical setting, algorithm, and/or model.
- ☒ A clear explanation of any assumptions.
- ☒ An analysis of the complexity (time, space, sample size) of any algorithm.

For any **theoretical claim**, check if you include:
- ❑ A clear statement of the claim.
- ❑ A complete proof of the claim.

For all **datasets** used, check if you include:
- ☒ The relevant statistics, such as number of examples.
- ☒ The details of train / validation / test splits.
- ☒ An explanation of any data that were excluded, and all pre-processing step.
- ☒ A link to a downloadable version of the dataset or simulation environment.
- ☒ For new data collected, a complete description of the data collection process, such as instructions to annotators and methods for quality control.

For all shared **code** related to this work, check if you include:
- ☒ Specification of dependencies.
- ☒ Training code.
- ☒ Evaluation code.
- ❑ (Pre-)trained model(s).
- ☒ README file includes table of results accompanied by precise command to run to produce those results.

For all reported **experimental results**, check if you include:
- ☒ The range of hyper-parameters considered, method to select the best hyper-parameter configuration, and specification of all hyper-parameters used to generate results.
- ☒ The exact number of training and evaluation runs.
- ☒ A clear definition of the specific measure or statistics used to report results.
- ☒ A description of results with central tendency (e.g. mean) & variation (e.g. error bars).
- ☒ The average runtime for each result, or estimated energy cost.
- ☒ A description of the computing infrastructure used.

Reproduced from: *www.cs.mcgill.ca/~jpineau/ReproducibilityChecklist-v2.0.pdf*

Figure S19: Reproducibility checklist