# OpenReview forum: "CableInspect-AD: An Expert-Annotated Anomaly Detection Dataset"
_NeurIPS.cc/2024/Datasets_and_Benchmarks_Track — NeurIPS 2024 Track Datasets and Benchmarks Poster_

### Official Review · Reviewer_U6dd · 2024-07-20
**The authors introduce CableInspect-AD, a high-quality visual anomaly detection dataset for robotic power line inspection.**

**Rating:** 6
**Confidence:** 3
**Correctness:** Yes
**Clarity:** Yes

**Review:**

1. The authors introduce CableInspect-AD, a high-quality visual anomaly detection dataset for robotic power line inspection.
2. The authors propose Enhanced-PatchCore to address the challenge of collecting various anomaly and nominal examples for setting detection thresholds.

**Strengths:**

1. The authors introduce CableInspect-AD, a high-quality visual anomaly detection dataset for robotic power line inspection.
2. The authors propose Enhanced-PatchCore to address the challenge of collecting various anomaly and nominal examples for setting detection thresholds.

**Additional Feedback:**

No

**Documentation:**

Yes

**Limitations:**

1. The dataset features a consistent and relatively uniform background, with no variations in color or brightness.
2. Is it feasible to address this issue using existing unsupervised anomaly detection methods?

**Opportunities For Improvement:**

1. The dataset features a consistent and relatively uniform background, with no variations in color or brightness.
2. Is it feasible to address this issue using existing unsupervised anomaly detection methods?
3. Please compare more recent zero/few-shot methods.

**Relation To Prior Work:**

Yes

**Summary And Contributions:**

This paper introduces CableInspect-AD, a high-quality dataset for visual anomaly detection in robotic power line inspection, created by Hydro-Québec experts. The authors enhance the PatchCore algorithm to handle limited labeled data and propose a comprehensive evaluation protocol. Despite promising results, the models struggle with all anomalies, highlighting the dataset's value as a challenging benchmark.

---

> ### Author Rebuttal · Authors · 2024-08-16
>
> We sincerely appreciate your thoughtful feedback and valuable insights. We have carefully considered your suggestions and updated the paper accordingly. We would like to respond to the points raised in detail below.
>
> > 1. The dataset features a consistent and relatively uniform background, with no variations in color or brightness.
>
> While our dataset does feature a consistent and uniform background, this choice was intentional. Our primary goal was to conduct a deep, focused study on a specific category of object, encompassing seven types of anomalies with varying severity levels, annotated at both the image and pixel levels by experts. This meticulous annotation process required substantial time and effort to ensure high quality. The uniform background was deliberately chosen to minimize distractions and external factors, allowing models to solely concentrate on detecting anomalies within the object. Moreover, real-world VAD applications often include preprocessing steps like region of interest extraction or other techniques for background removal. For example, our robots used for powerline inspection are equipped with apparatuses that control environmental factors, making background uniformity less of a concern in practical scenarios. Despite the uniformity, our models still face challenges in detecting all anomalies, which underscores the dataset's complexity and rigorous testing capability. While exploring additional variations was beyond the scope of this study, we believe our dataset and approach provide valuable insights into the effectiveness of models in real-world contexts by demonstrating how models perform under these controlled, yet challenging conditions.
>
> > 2. Is it feasible to address this issue using existing unsupervised anomaly detection methods?
>
> Addressing this issue with existing unsupervised anomaly detection methods presents significant challenges. Unsupervised methods typically rely solely on nominal images during training, yet they require both nominal and anomalous images to establish an effective detection threshold. Setting this threshold is particularly difficult without labeled data, as evidenced by recent studies like AnomalyGPT [1]. Although adapting unsupervised methods to our specific context is theoretically possible, it would require significant modifications and a number of labeled examples, which are beyond the scope of our current work. A lack of diverse anomalies can lead to overfitting and unreliable performance in practice. Our Enhanced-PatchCore method addresses this limitation by providing a more robust approach that relies only on nominal images in the train set for thresholding, making it more suitable for practical scenarios. Additionally, we evaluated zero-shot settings using Vision Language Models (VLMs) as an alternative approach.
>
> >3. Please compare more recent zero/few-shot methods.
>
> In our paper, we compare five recent open conversational VLMs of various sizes [2, 3], ranging from LLaVA-1.5-7B to CogVLM2-19B released in May 2024 for the zero-shot setting. We also include WinCLIP [4] for this setting, which is a pioneer model using CLIP for VAD. For few-shot AD, we extended the popular PatchCore algorithm to automatically set the threshold using only two or more nominal images. In addition, [5] shows the potential of PatchCore in few-shot scenarios. While we considered several methods as baselines, we did not include methods that needed a considerable amount of tweaks and modifications. For example, we did not include Segment Any Anomaly [6] because it has several prompting modules that require heavy application specific prompt engineering. For zero-shot, we intentionally excluded closed-source models like GPT-4V due to their proprietary nature and limited API access, which limit transparency and reproducibility. Our focus remains on open models to ensure our comparisons are accessible and replicable within the research community. If there are specific zero-shot or few-shot models you would like us to include, please let us know, and we will consider adapting and incorporating them. We also hope that our dataset can serve as a benchmark for future development of more robust zero- and few-shot approaches.
>
> References:
>
> [1] Gu, Z., Zhu, B., Zhu, G., Chen, Y., Tang, M. and Wang, J., 2024, March. Anomalygpt: Detecting industrial anomalies using large vision-language models. In Proceedings of the AAAI Conference on Artificial Intelligence (Vol. 38, No. 3, pp. 1932-1940).
>
> [2] Wang, W., Lv, Q., Yu, W., Hong, W., Qi, J., Wang, Y., Ji, J., Yang, Z., Zhao, L., Song, X. and Xu, J., 2023. Cogvlm: Visual expert for pretrained language models. arXiv preprint arXiv:2311.03079.
>
> [3] Liu, H., Li, C., Li, Y. and Lee, Y.J., 2024. Improved baselines with visual instruction tuning. In Proceedings of the IEEE/CVF Conference on Computer Vision and Pattern Recognition (pp. 26296-26306).
>
> [4] Jeong, J., Zou, Y., Kim, T., Zhang, D., Ravichandran, A. and Dabeer, O., 2023. Winclip: Zero-/few-shot anomaly classification and segmentation. In Proceedings of the IEEE/CVF Conference on Computer Vision and Pattern Recognition (pp. 19606-19616).
>
> [5] Santos, J., Tran, T. and Rippel, O., 2023. Optimizing patchcore for few/many-shot anomaly detection. arXiv preprint arXiv:2307.10792.
>
> [6] Cao, Y., Xu, X., Sun, C., Cheng, Y., Du, Z., Gao, L. and Shen, W., 2023. Segment any anomaly without training via hybrid prompt regularization. arXiv preprint arXiv:2305.10724.

---

> > ### Comment · Reviewer_U6dd · 2024-08-22
> > **Thanks and more questions.**
> >
> > Thank you for your response. I have the additional concerns:
> >
> > 1. If the author intentionally uses a uniform background, it significantly reduces the difficulty of anomaly detection, particularly in anomaly localization. This is because existing methods are often hindered by background false positives.
> >
> > 2. For optical cable inspection on a production line, why use VLM for anomaly detection? Typically, VLM's detection efficiency is concerning. It is recommended that the author provide detection speed or FPS metrics.
> >
> > 3. I believe that using unsupervised anomaly detection methods in this scenario with a single background is feasible. The distinction between normal and anomalous samples is substantial. It is suggested that the author compare several classic unsupervised anomaly detection methods such as RD4AD, SimpleNet, UniAD, and EfficientAD, which are likely to have a speed advantage over VLM.

---

> > ### Author Response · Authors · 2024-08-23
> > **Response to additional questions**
> >
> > Thank you for your response. We appreciate the opportunity to address your additional concerns. We will update the paper to include the inference speed analysis and emphasize the complexity and diversity of the anomalies in our dataset.
> >
> > 1. **Challenging dataset**: While a uniform background might simplify anomaly detection, our dataset still presents significant challenges. The best detection and segmentation metrics—F1-score of 0.77 ± 0.02 (Table 1) and AUPRO of 0.53 ± 0.08 (Section 6 - Anomaly Segmentation)—reveal that even with these conditions, the anomalies remain difficult to detect and especially to localize accurately. This underscores the complexity of the anomalies in our dataset which is our primary contribution: a collection of detailed and nuanced, rare-to-find anomalies with thorough annotations. These anomalies range from subtle scratches to major structural defects in cables, with varying levels of natural wear. Our findings show that despite the uniform background, current methods struggle with these real-world challenges, highlighting the limitations of existing approaches. This underscores the need for more advanced algorithms capable of detecting subtle defects and handling natural wear.
> >
> > 2. **Justification for including VLMs as a baseline**: VLMs can offer several advantages, such as adaptability to various cases without needing specific training, multimodal integration of visual and textual data, and zero-shot capabilities that mitigate sensitivity to dataset contamination. Our experiments revealed that VLMs can outperform Enhanced-PatchCore in certain domain-specific scenarios, challenging initial assumptions about their efficiency.
> >
> >    Regarding speed, we acknowledge that VLMs typically exhibit lower detection rates compared to traditional methods, with CogVLM-17B achieving 2.3 FPS versus Enhanced-PatchCore’s 5.68 FPS, using a batch size of 1 on A100 machine. For high-speed applications like real-time production monitoring, Enhanced-PatchCore may be more suitable. However, VLMs may offer significant advantages for post-process inspections, which is the case in powerline inspection, where adaptability and interpretability are crucial. Optimizing VLM processing speed and exploring hybrid approaches that leverage the strengths of both VLMs and traditional methods could be promising directions for future research for practical applications. However, we want to emphasize that the goal of this work is not to recommend one method over another but to provide a challenging dataset with promising baseline models. This work is intended to stimulate further research and development in anomaly detection, highlighting the complexities involved and encouraging the exploration of both traditional and novel methods.
> >
> > 3. **Limitations of unsupervised AD methods**: Classical unsupervised anomaly detection methods such as RD4AD, SimpleNet, UniAD, and EfficientAD typically require labeled data—nominal images during training and both nominal and anomalous images during validation to establish detection thresholds [1,3,4]. This requirement makes these methods impractical for real-world applications due to the significant time, cost, and expertise needed for annotation. Consequently, we have not included these models as baselines in our study, as the annotation overhead often outweighs any potential speed advantages. While we have already discussed the limitations of unsupervised methods in the related work section, we will expand our analysis to include these specific models.
> >
> >    Recent research is increasingly exploring zero-shot and few-shot approaches for anomaly detection, which bypass the need for extensive labeled data by not requiring custom models to be trained for each specific task. For instance, methods like WinCLIP [2] have demonstrated superior performance compared to traditional approaches on popular anomaly detection benchmarks. Given these advancements, we opted to focus on few-shot and zero-shot methods in our work. These approaches are more practical, do not require extensive labeled datasets, and are in line with the current trends and future direction of the field.
> >
> >    The wide range of anomalies in our dataset makes distinguishing between nominal and anomalous samples challenging, as detailed in our response to your first point. This complexity underscores the relevance of our dataset.
> >
> > Reference:
> >
> > [1] Yoon, Jinsung, et al. "Self-supervise, refine, repeat: Improving unsupervised anomaly detection." arXiv:2106.06115 (2021).
> >
> > [2] Jeong, J., Zou, Y., Kim, T., Zhang, D., Ravichandran, A. and Dabeer, O., 2023. Winclip: Zero-/few-shot anomaly classification and segmentation, CVPR.
> >
> > [3] Yu, Jongmin, et al. "Normality-calibrated autoencoder for unsupervised anomaly detection on data contamination."  arXiv:2110.14825 (2021).
> >
> > [4] Gu, Z., Zhu, B., Zhu, G., Chen, Y., Tang, M. and Wang, J., 2024, March. Anomalygpt: Detecting industrial anomalies using large vision-language models, AAAI.

---

> > > ### Comment · Reviewer_U6dd · 2024-08-26
> > > **Response to Rebuttal**
> > >
> > > Thank you for your response. However, I have some differing opinions on a few points:
> > >
> > > 1. The inference speed of the selected VLM and PatchCore-based methods is significantly limited. Given that power lines are typically quite long, slow inference speeds may not meet the requirements.
> > > 2. Only WinCLIP, a single zero-shot and few-shot comparison method, was selected, which is not sufficiently convincing. Please compare it with the latest zero-shot and few-shot methods such as MUSC and APRIL-GAN.

---

> > > > ### Author Response · Authors · 2024-08-27
> > > > **Response to additional points**
> > > >
> > > > Thank you for continuing the discussion and providing valuable feedback on our work.
> > > >
> > > > 1. We appreciate the reviewer's concern about inference speed. We would like to emphasize that the goal of our work is to contribute a real-world dataset to help identify algorithms or models that can achieve a high detection rate of anomalies. Once a satisfying approach is found, we can apply optimization to aim for real-time inference if needed as a next step. The focus of this work is on assessing the performance of zero- and few-shot anomaly detection methods to establish baseline performance, which is the most critical aspect when introducing a new dataset.
> > > >
> > > >    We believe that the inference speed is context-dependent and should be considered when adapting methods for deployment on specific hardware in particular scenarios. It is important to note that according to the experts at Hydro-Québec, constant monitoring of the power lines is not necessary; these operations would be conducted periodically and would not involve inspecting all lines simultaneously. Furthermore, real-time detection is not always a requirement. Post-processing, such as overnight data inspection is acceptable by Hydro-Québec, providing a large time window for inference.
> > > >
> > > > 2. We understand the reviewer’s suggestion to include comparisons with additional methods such as MuSc and APRIL-GAN, and we value this feedback. However, our decision to focus on WinCLIP was based on careful consideration of its alignment with the unique challenges posed by our dataset and real-world applicability.
> > > >
> > > >    For MuSc, although it is claimed to be a zero-shot method, it still requires prior knowledge from a test set [1], which is not feasible in real-world scenarios, especially in power line inspection. In addition, the method assumes the test set contains abundant information on both normal and abnormal cues, which is not suitable in scenarios where only nominal images are available. Due to these limitations, we did not consider MuSc as an appropriate benchmark method for our use case.
> > > >
> > > >    APRIL-GAN, while it achieves strong results in selected scenarios, requires an additional training phase, which is resource-intensive to be trained and evaluated on our dataset. Furthermore, WinCLIP either matches or outperforms APRIL-GAN in similar contexts [2], making it a more relevant choice for initial benchmarking.
> > > >
> > > >    Therefore, we prioritize WinCLIP over the aforementioned two methods. We have also included a variant of PatchCore in a few-shot setting. These methods have been widely adopted in related research, which aligns with the goal of benchmarking our dataset using a recognized standard. We would like to reiterate that we are interested in methods that either employ a pure zero-shot setup without any prior knowledge required, or a few-shot setup relying exclusively on nominal samples (as demonstrated by our proposed Enhanced-PatchCore). Nevertheless, we will update the related work section of the paper to clearly explain why we did not consider MuSc and APRIL-GAN as relevant baselines in our scenario.
> > > >
> > > > Thank you again for your helpful feedback. We hope that our explanations clarify our decisions, and we're looking forward to seeing how our dataset might be used in future research, including the exploration of inference speed optimizations and the evaluation of additional state-of-the-art models.
> > > >
> > > > If you have any further questions or concerns, or if there are other aspects you would like us to clarify, please let us know.
> > > >
> > > > Reference:
> > > >
> > > > [1] Li, Xurui, et al. "Musc: Zero-shot industrial anomaly classification and segmentation with mutual scoring of the unlabeled images." arXiv preprint arXiv:2401.16753 (2024).
> > > >
> > > > [2] Chen, Xuhai, Yue Han, and Jiangning Zhang. "APRIL-GAN: A Zero-/Few-Shot Anomaly Classification and Segmentation Method for CVPR 2023 VAND Workshop Challenge Tracks 1&2: 1st Place on Zero-shot AD and 4th Place on Few-shot AD." arXiv preprint arXiv:2305.17382 (2023).

---

> > > > > ### Comment · Reviewer_U6dd · 2024-08-29
> > > > > **Thanks**
> > > > >
> > > > > Thanks to the author for the rebuttal. Addresses most of my concerns. So I upgraded the score.

---

> > > > > > ### Author Response · Authors · 2024-08-29
> > > > > > **Thank you**
> > > > > >
> > > > > > We wanted to express our sincere gratitude for your thoughtful review and for increasing the score of our paper. We greatly appreciate the time and effort you put into evaluating our work. Thank you once again.

---

### Official Review · Reviewer_hjBg · 2024-07-23
**A real-world cable dataset for anomaly detection**

**Rating:** 5
**Confidence:** 4
**Correctness:** The dataset has been constructed in a…
**Clarity:** The paper is written well.

**Review:**

The authors propose a real-world cable dataset, which is suitable for cable anomaly detection scenarios, but has limitations in other scenarios. Enhanced-PatchCore is proposed to set thresholds using only few normal images. To get rid of the dependence on training images, the authors use conversational VLMs for zero-shot anomaly detection, achieving promising results on this dataset.

**Strengths:**

1. The authors propose a real-world cable dataset. The high-resolution images and rare multi-scale anomalies are suitable for cable anomaly detection scenarios.
2. Enhanced-PatchCore is proposed to set image-level classification thresholds using only few normal images. The results of the four proposed threshold strategies are analyzed comprehensively.
3. The authors try more VLMs on the zero-shot industrial anomaly detection task, and obtain satisfactory segmentation performance and a low FPR.

**Additional Feedback:**

None

**Documentation:**

The authors provide the data collection and organization details, licensing and ethical and responsible use. The authors provide a download link for the dataset, but it appears that the website is temporarily unavailable.

**Limitations:**

The limitations of this dataset have been declared by the authors. Other possible limitations are written in the Opportunities For Improvement.

**Opportunities For Improvement:**

1. There is only the cable category in this dataset, which may not apply to the broad scenarios.
2. There are some other conversational VLMs papers [1][2] for anomaly detection that need to be discussed. Whether the attempt in this paper is the first of its kind is questionable.
3. The Enhanced-PatchCore proposed in this paper can only determine the image-level threshold. Is there a strategy to determine a pixel-level threshold？
4. The authors report FPR in the experiments, but in the anomaly detection scenario, FNR deserves more attention. It is more beneficial to report FNR in the experiments.

[1] Gu Z, Zhu B, Zhu G, et al. Anomalygpt: Detecting industrial anomalies using large vision-language models[C]//Proceedings of the AAAI Conference on Artificial Intelligence. 2024, 38(3): 1932-1940.

[2] Cao Y, Xu X, Sun C, et al. Towards generic anomaly detection and understanding: Large-scale visual-linguistic model (gpt-4v) takes the lead[J]. arXiv preprint arXiv:2311.02782, 2023.

**Relation To Prior Work:**

See the Opportunities For Improvement.

**Summary And Contributions:**

The authors propose a real-world cable dataset, which contains the high-resolution images with rare multi-scale anomalies. This dataset provides image-level, pixel-level, and bounding box annotations. Enhanced-PatchCore is proposed to set thresholds using only few normal images. This method is suitable for scenarios where labeled images are unavailable. To get rid of the dependence on training images, the authors use conversational VLMs for zero-shot anomaly detection, achieving promising results on the cable dataset.

---

> ### Author Rebuttal · Authors · 2024-08-16
>
> We sincerely appreciate your thoughtful feedback and the opportunity to address the points raised. We have updated the paper to incorporate your suggestions and address all of your comments. We provide more details below:
>
> >1. There is only the cable category in this dataset, which may not apply to the broad scenarios.
>
> While our focus on power line cables may seem limited, our dataset's strength lies in its diverse seven anomaly types with three severity levels. This allows for a more in-depth evaluation within the targeted domain. Our dataset was designed by experts to address the challenges of a real-world application: power line cable inspection using robots like the LineRanger, the LineScout or the LineDrone.
>
> While broad datasets are valuable for assessing generalization across diverse categories, they may not fully capture the intricacies required for real-world applications. They often include fewer types or grades of anomalies per category, which can limit their effectiveness in applications where precision and reliability are crucial. For example, models like PatchCore have achieved near-100% accuracy on MVTec AD, yet they struggle to maintain this level of performance on our dataset.
>
> The methodologies and insights derived from our focused study are not only robust but also adaptable to a wide range of anomaly detection scenarios. For instance, our experiments demonstrate that VLMs can be effectively utilized for zero-shot VAD tasks. However, we also find that no current model performs well across all anomaly types, particularly when detecting light-grade anomalies. This finding reveals the limitations of current models and provides a valuable direction for future research aimed at enhancing model performance in specialized applications.
>
> > 2. There are some other conversational VLMs papers [1][2] for anomaly detection that need to be discussed. Whether the attempt in this paper is the first of its kind is questionable.
>
> We acknowledge the importance of discussing relevant works like AnomalyGPT [1] and the study using GPT-4V for VAD [2], which we have already cited in our related work section. Our work, however, differs in several key aspects. First, GPT-4V is a proprietary model with limitations in terms of cost, compute and privacy, whereas our paper specifically focuses on open conversational VLMs. Our decision to use open models is intentional, prioritizing greater transparency, accessibility, and reproducibility in practical applications. Additionally, the work using GPT-4V for VAD [2] presents exclusively qualitative results, while our work presents quantitative evaluation of conversational VLMs. Second, AnomalyGPT is a conversational VLM fine-tuned for VAD tasks, requiring a set of nominal and simulated anomalous images. In contrast, our work focuses on zero-shot scenarios with the aim of eliminating the need for data annotation and augmentation, which can be costly in VAD applications. By demonstrating the zero-shot potential of open solutions in this domain, we offer a complementary perspective that is both adaptable and extendable by the research community. To our knowledge, there have been no prior studies evaluating the performance of multiple VLMs in this context.
>
> > 3. The Enhanced-PatchCore proposed in this paper can only determine the image-level threshold. Is there a strategy to determine a pixel-level threshold？
>
> Enhanced-PatchCore operates at both the image and pixel levels in a similar manner. Specifically, at the pixel level, the anomaly score for a pixel at coordinates $(i, j)$ in the image is computed using the following equation:
>
> $$
> S(X_{i,j}) := \min_{e' \in \mathcal{M} \setminus \mathcal{P}(X)} d(e_{i, j}, e'),
> $$
>
> where $d$ represents the Euclidean distance between the patch embedding covering the pixel of interest $e_{i, j}$ and the patch embeddings stored in the memory bank $e' \in \mathcal{M}$, excluding those from the embedding set $\mathcal{P}(X)$ corresponding to the embeddings of the image containing the pixel of interest.  The thresholding strategies used at image level can then be applied to the pixel scores distribution. In the paper, our anomaly segmentation results (Section 6, Anomaly Segmentation) are obtained using this approach.
>
> > 4. The authors report FPR in the experiments, but in the anomaly detection scenario, FNR deserves more attention. It is more beneficial to report FNR in the experiments.
>
> We fully agree that FNR is crucial, especially in scenarios where missing anomalies can have serious consequences. Since FNR can be defined as 1- Recall, we considered this metric to be investigated indirectly in our study through the observations on Recall. For instance, a high recall indicates that the model correctly identifies most of the anomalies, implying a low FNR. In this work, we aimed to provide a balanced evaluation of the models by including FPR, precision, recall, and other relevant metrics. Therefore, to provide a more thorough evaluation, we have expanded our results analysis to include FNR alongside the other metrics.
>
> We sincerely thank the reviewer for their thoughtful feedback. Our work, developed with domain experts on a real-world problem, supports open science (see checklist and datasheet) and bridges academic research with practical applications. We emphasize the dataset's complexity and detailed annotations as key contributions. We've addressed all concerns and believe the revisions strengthen the paper. We hope these meet your expectations and positively impact your evaluation. Please let us know if you have any further questions.
>
> References:
>
> [1] Gu, Z., Zhu, B., Zhu, G., Chen, Y., Tang, M. and Wang, J., 2024, March. Anomalygpt: Detecting industrial anomalies using large vision-language models. AAAI 2024.
>
> [2] Cao, Y., Xu, X., Sun, C., Huang, X. and Shen, W., 2023. Towards generic anomaly detection and understanding: Large-scale visual-linguistic model (gpt-4v) takes the lead. arXiv preprint arXiv:2311.02782.

---

> > ### Comment · Reviewer_hjBg · 2024-08-23
> > **Response to Rebuttal**
> >
> > Thanks to the author for solving most of my problems, I understand the importance of CableInspect-AD datasets in the industrial field.
> >
> > My question in **Q3** is whether the Enhanced PatchCore has a strategy to determine a pixel-level **threshold**. Only the AUPRO of heat maps are given in Section 6 Anomaly Segmentation. I noticed that the pixel-level prediction heatmap (left) in Figure 4 seems to be segmented by a threshold, and it would be better if quantitative metrics could be provided.

---

> > > ### Author Response · Authors · 2024-08-23
> > > **Response: Clarification on Pixel-Level Thresholding**
> > >
> > > Thank you for your thoughtful response and for following-up on Q3. We're glad that our previous responses have addressed most of your concerns.
> > >
> > > We assume the question raised refers to Figure 7, which is the one that illustrates anomaly segmentation for Enhanced-PatchCore.
> > >
> > > Regarding your question about the pixel-level thresholding strategy for Enhanced-PatchCore, we want to clarify that we apply thresholding not only at the image level but also at the pixel level to generate anomaly segmentation maps. You correctly identified that the pixel-level prediction heatmap in Figure 7 was segmented using a thresholding strategy. However, we acknowledge that the initial draft did not explicitly include the corresponding quantitative metrics.
> > > The segmentation results shown in Figure 7 were obtained using a max thresholding strategy as described in Figure S3 in the Supplementary material. This approach selects the maximum anomaly score from the empirical distribution of the training data as the threshold, based on the assumption that the training data contains only nominal images. Ideally, this threshold should be lower than the scores associated with anomalies in the test set. The corresponding pixel-level metric, the Pixel-wise Overlap (PRO) score, averaged across all cables and folds, is 0.28 ± 0.09. As you can see, this score is relatively low, indicating that there is room for improvement in the segmentation task. We agree that including these quantitative metrics enhances our discussion in Section 6 on Anomaly Segmentation, and we have updated the paper to reflect this.
> > >
> > > Thank you again for your helpful feedback, which has guided us in making these valuable updates to the paper.

---

> > ### Author Response · Authors · 2024-08-29
> > **Follow-Up**
> >
> > We sincerely appreciate your ongoing engagement and the constructive feedback you have provided throughout this discussion period. As the discussion phase is drawing to a close tomorrow, we wanted to take this opportunity to address any remaining concerns.
> >
> > We have carefully considered and incorporated your suggestions, especially regarding the pixel-level thresholding strategy in Enhanced-PatchCore. We provided additional clarification and quantitative metrics to enhance the transparency and completeness of our anomaly segmentation analysis. We believe they have strengthened the overall quality and robustness of our paper, in line with your suggestions.
> >
> > We would be deeply grateful if you could revisit our submission in light of these updates. Please let us know if there are any further aspects you would like us to discuss.

---

### Official Review · Reviewer_jyZc · 2024-07-25
**Expert annotations and real-world applications represent the gold standard of ML benchmarks**

**Rating:** 8
**Confidence:** 4

**Review:**

Pros:
- Provides an incredible dataset with a variety of anomaly types for power line inspection.
- Introduces an Enhanced PatchCore method grounded in real-scenarios where training information is nominal.
- Thorough description of performance tradeoffs.

Cons:
- Data not yet in FAIR repo with persistent identifier
- Expert annotated ground truth, while excellent, could use some additional discussion about potential for bias or differing expert opinions (or how that was mitigated)

**Strengths:**

Strengths:
1. This submission includes a thorough, unique ground truth anomaly detection dataset in an important VAD application space. Their expert labels are included at image levels, in bounding boxes, and per pixel for the first video on each cable. The sheer thoroughness of the ground truth in this dataset is honestly unprecedented in this and many fields--especially since traditional research in this space has been unsupervised. The opportunities for reproducible validation on this dataset are astounding.

2. The benchmark portion is clear in which methods it is testing against and how they are run, but is particularly impressive in its thorough efforts to simulate how these tools would be used in a realistic VAD pipeline.

**Additional Feedback:**

Please, as long as there are no data stakeholder issues with it, host your data in a FAIR data repository with a persistent identifier and standard metadata. There are many excellent ones listed on https://fairsharing.org/databases/ or https://www.re3data.org/ in both domain specific and general spheres.

Other than that this was a pleasure to read.

**Clarity:**

This paper is very clear and methodically laid out. I was particularly impressed by the methodical progression of the exposition on performance results, and how each step in the benchmark connected back to real world scenarios.

**Correctness:**

The discussion in this paper, to the best of my knowledge, was absolutely correct. The authors included thorough explanations for each design and methodological choice both in the data set and the benchmark and took great care to connect it to reality.

**Documentation:**

In terms of documentation, the paper covers a lot of ground effectively and efficiently. The reasoning behind and implementation of enhanced-patch core were clear, and the description of the dataset gave enough detail in the main body of the paper as to not be wholly reliant on the supplementary material.

The website for the project provides a good introduction, and the code is well described in github. However, this work does have one key issue, in that the dataset is hosted on the website rather than in a FAIR data repository and does not have an associated persistent identifier (such as a DOI) or obvious metadata management.

I recommend that the data creators host their project in a FAIR data repository unless data stakeholders (Hydro-Quebec experts, perhaps?) have stipulated that it cannot be hosted elsewhere. A good option for this project would be Zenodo, since the dataset size limit on Zenodo is 50Gb (which this is certainly under at 12 Gb) and Zenodo links seamlessly with Github codebases.

**Ethics:**

No obvious ethics flags in this dataset

**Limitations:**

The limitations section in this paper was excellent because it immediately highlighted a key concern in many anomaly detection datasets--anomaly ratios--as well as a potential solution. It also was specific about some of the anomalies that couldn't be captured by the dataset in spite of the many highlighted efforts to mimic reality.

I was also impressed that the authors included a reminder that ML in robotic power line inspection can have critical operational consequences; this highlighted the importance of potential future studies on trustworthiness for VAD in this application space.

**Opportunities For Improvement:**

This was a terribly impressive paper. I mention this in documentation below, but the biggest issue with it is that the dataset is not stored in a FAIR repository, it is downloadable from the dataset website. This means it is open, but doesn't have a persistent identifier or added searchability/findability in an appropriate repository.

One other thing that could potentially be addressed is an issue common to expert labelled ground truth--how was bias or individual opinion mitigated in the labelling process? Were multiple experts involved, was there any quantification or validation for their labelling process? It could be worth discussing briefly whether expert opinion could have any uncertainty.

The only other thing I might mention (and this is absolutely just a little extra bit of feedback because the quality is already so high) is that there are opportunities for improved accessibility. Figure 2 and Figure 3 both use color-based identification that might be difficult for researchers with CVD to interpret.

**Relation To Prior Work:**

Section 2 succinctly and thoroughly outlines available industrial datasets related to power line inspection and visual anomaly detection, and highlights where the CableInspect-AD fills unique gaps in state of the data. It also gives a brief but comprehensive review of reconstruction based and feature embedding based approaches for VAD in industrial settings and describes large models and VLMs in the context of VAD.

**Summary And Contributions:**

This work introduces a large gold-standard dataset of expert annotated power cables for robotic visual anomaly detection (VAD) tasks. Care was taken to include the nuances and complexities of real cable anomalies--typically missing from existing VAD datasets in this space. The benchmark on this dataset explores the use of vision language models for the VAD task, but does so using performance evaluation that reflects the nuances of operational performance (over a simple supervised pipeline with basic accuracy/precision/recall statistics).

---

> ### Author Rebuttal · Authors · 2024-08-16
>
> We sincerely appreciate your thoughtful and detailed feedback. We are thrilled that you found our dataset and Enhanced PatchCore method both impressive and valuable. Your recognition of the thoroughness of our ground truth annotations and the practical relevance of our benchmark is highly encouraging.
>
> We agree that a significant limitation of current machine learning benchmarks is their lack of real-world applicability. State-of-the-art performance on academic datasets often fails to translate to practical scenarios, making it challenging to adopt these methods in industry. We hope our work helps bridge this gap by offering a more practical and industry-relevant benchmark.
>
> Regarding the dataset, you are correct that Hydro-Québec currently prefers to manage its hosting. However, we recognize the importance of compliance with FAIR principles and are discussing better hosting options, and to make the platform FAIR compliant.
>
> We also appreciate your suggestion regarding color vision deficiency (CVD) accessibility in the figures. In the camera-ready version, we have adjusted the figures to be more CVD-friendly.
>
> The dataset was annotated by at least four experts who first developed and agreed on guidelines to establish a clear annotation framework. The dataset then underwent five iterative rounds of review and feedback, allowing the experts to reach a consensus. This process ensured that the final version is both reliable and reflective of real-world conditions. While very light anomalies, such as light deposits and scratches, might have been missed, the experts agreed these are not critical, as they would not require immediate repair in a real-world scenario and might even go undetected by field experts. All mild and severe cases were thoroughly annotated. We did not quantify the annotation process’ performance, as it was conducted in a consensus-driven, iterative manner until an agreement was reached.
>
> Thank you again for your constructive comments. We believe these updates will address your suggestions and further strengthen the paper.

---

> > ### Comment · Area_Chair_Z5R8 · 2024-08-29
> >
> > Can Reviewer jyZc please comment on the authors' rebuttal? Thank you.

---

### Author Rebuttal · Authors · 2024-08-16

We sincerely thank the reviewers for their thoughtful feedback on our paper. We are pleased that our work, particularly the introduction of a high-quality real-world dataset and the Enhanced-PatchCore method, was acknowledged as significant. We greatly appreciate Reviewer jyZc’s positive evaluation and have carefully addressed the constructive feedback from Reviewers hjBg and U6dd to strengthen our paper.

Across the reviews, several key strengths of our paper were highlighted:
1. Unique dataset: The introduction of CableInspect-AD, an expert-annotated dataset tailored for visual anomaly detection (VAD) in power line cable inspection, featuring detailed annotations at multiple levels (image, bounding box, pixel) and rare multi-scale anomalies.
2. Enhanced-PatchCore: The proposal of the Enhanced-PatchCore method, which effectively sets detection thresholds using minimal nominal examples, making it ideal for scenarios with limited labeled data.
3. Comprehensive evaluation: A thorough and realistic evaluation process that benchmarks performance while accounting for the nuances of real-world operational environments.
4. Clarity and documentation: The paper’s clarity, documentation, and well-executed discussion of limitations and relation to prior work, consistently praised across all reviews.

We acknowledge the concerns raised by the reviewers and have addressed them comprehensively. Below is a summary of the changes made for the camera-ready version (detailed responses to each reviewer are provided individually):
1. FAIR Data Repository: We have initiated discussions with Hydro-Québec about better hosting options and to make their platform compliant with the FAIR principles to enhance its accessibility and impact within the community.
2. Details on Expert Annotations: We have expanded our discussion on the expert annotations, offering greater transparency on how the associated challenges were addressed.
3. VLM Comparisons and FNR Reporting: Our work already references the mentioned methods (i.e. GPT-4V [1,2] and AnomalyGPT [3]). We have expanded the discussion of recent conversational VLMs for anomaly detection and clarified the novelty of our approach within this context. While FNR is an important metric in VAD, we initially did not discuss it explicitly because FNR is indirectly reflected through recall [FNR = 1 - Recall]. However, to provide a more thorough evaluation, we have expanded our analysis to include FNR alongside FPR, precision, recall, and other relevant metrics.
4. Dataset Background Uniformity: The uniform background minimizes distractions, letting models focus on detecting object anomalies. In real-world VAD applications, preprocessing like background removal is common. For example, robots used for powerline inspection can be equipped with apparatuses that control environmental factors, making background uniformity less of a concern in practical scenarios. Despite the uniformity, the models still struggle with some anomalies, highlighting the dataset's complexity and importance.
5. Other updates: We have added more details on differentiating our dataset from other datasets with broad categories, and on unsupervised and zero-shot methods for VAD. We extended the Enhanced-PatchCore section to explain how it works for pixel-level prediction and adjusted the figures to make them CVD friendly.

Overall, the strong endorsement of our dataset's value, its potential for reproducible research, and its real-world applicability underscores the significant impact our work can have on the community. Given the importance of open science and the extensive efforts invested in creating this expert-annotated dataset, we hope the reviewers will reconsider their scores to further support the dissemination and development of practical, real-world datasets in our field.

Thank you once again for your invaluable feedback. We believe these revisions will strengthen our paper and enhance its contribution to the community.

References:

[1] OpenAI (2023). GPT-4V(ision) system card. OpenAI. 25 September. (Accessed: 30 November 2023)

[2] Cao, Y., Xu, X., Sun, C., Huang, X. and Shen, W., 2023. Towards generic anomaly detection and understanding: Large-scale visual-linguistic model (gpt-4v) takes the lead. arXiv preprint arXiv:2311.02782.

[3] Gu, Z., Zhu, B., Zhu, G., Chen, Y., Tang, M. and Wang, J., 2024, March. Anomalygpt: Detecting industrial anomalies using large vision-language models. In Proceedings of the AAAI Conference on Artificial Intelligence (Vol. 38, No. 3, pp. 1932-1940).

[4] Bergmann, P., Fauser, M., Sattlegger, D. and Steger, C., 2019. MVTec AD--A comprehensive real-world dataset for unsupervised anomaly detection. In Proceedings of the IEEE/CVF conference on computer vision and pattern recognition (pp. 9592-9600).

[5] Zou, Y., Jeong, J., Pemula, L., Zhang, D. and Dabeer, O., 2022, October. Spot-the-difference self-supervised pre-training for anomaly detection and segmentation. In European Conference on Computer Vision (pp. 392-408). Cham: Springer Nature Switzerland.

---

### Decision · Program_Chairs · 2024-09-26

**Decision:**

Accept (Poster)

**Comment:**

This paper received 1 negative review and 2 positive reviews, with the average rating of 6.33. The average rating places this paper above the acceptance threshold.